# Ultrafast photochemistry produces superbright short-wave infrared dots for low-dose in vivo imaging

Harrisson D. A. Santos[1,2,12], Irene Zabala Gutiérrez [3,12], Yingli Shen[1,12], José Lifante[4,5,12], Erving Ximendes [1,4,12], Marco Laurenti [3,4], Diego Méndez-González[3], Sonia Melle [6], Oscar G. Calderón [6], Enrique López Cabarcos[3], Nuria Fernández [4,5], Irene Chaves-Coira[7], Daniel Lucena-Agell [10], Luis Monge[4,5], Mark D. Mackenzie[8], José Marqués-Hueso[9], Callum M. S. Jones [9], Carlos Jacinto[2], Blanca del Rosal[11], Ajoy K. Kar[8], Jorge Rubio-Retama [3,4✉] & Daniel Jaque [1,4✉]

Optical probes operating in the second near-infrared window (NIR-II, 1,000-1,700 nm), where tissues are highly transparent, have expanded the applicability of fluorescence in the biomedical field. NIR-II fluorescence enables deep-tissue imaging with micrometric resolution in animal models, but is limited by the low brightness of NIR-II probes, which prevents imaging at low excitation intensities and fluorophore concentrations. Here, we present a new generation of probes ($Ag_2S$ superdots) derived from chemically synthesized $Ag_2S$ dots, on which a protective shell is grown by femtosecond laser irradiation. This shell reduces the structural defects, causing an 80-fold enhancement of the quantum yield. PEGylated $Ag_2S$ superdots enable deep-tissue in vivo imaging at low excitation intensities ($<10$ mW cm$^{-2}$) and doses ($<0.5$ mg kg$^{-1}$), emerging as unrivaled contrast agents for NIR-II preclinical bioimaging. These results establish an approach for developing superbright NIR-II contrast agents based on the synergy between chemical synthesis and ultrafast laser processing.

[1] Fluorescence Imaging Group, Departamento de Física de Materiales, Facultad de Ciencias, Universidad Autónoma de Madrid, Madrid 28049, Spain. [2] Group of Nano-Photonics and Imaging, Instituto de Física, Universidade Federal de Alagoas, Maceió-AL 57072-900, Brazil. [3] Departamento de Química en Ciencias Farmacéuticas, Universidad Complutense de Madrid, Madrid 28040, Spain. [4] Nanobiology Group, Instituto Ramón y Cajal de Investigación Sanitaria, Hospital Ramón y Cajal, 28034 Madrid, Spain. [5] Fluorescence Imaging Group, Departamento de Fisiología, Facultad de Medicina, Universidad Autónoma de Madrid, Madrid 28029, Spain. [6] Department of Optics, Complutense University of Madrid, 28037 Madrid, Spain. [7] Departament of Anatomy, Histology and Neuroscience, Facultad de Medicina, Universidad Autónoma de Madrid, Madrid 28029, Spain. [8] Institute of Photonics and Quantum Sciences (IPaQS), School of Engineering and Physical Sciences, Heriot-Watt University, Edinburgh EH14 4AS, UK. [9] Institute of Sensors, Signals and Systems (ISSS), School of Engineering & Physical Sciences (EPS), Heriot-Watt University, Edinburgh EH14 4AS, UK. [10] Chemical and Physical Biology, Centro de Investigaciones Biologicas, Consejo Superior de Investigaciones Cientificas CIB–CSIC, Madrid 28040, Spain. [11] Centre for Micro-Photonics, Faculty of Science, Engineering and Technology, Swinburne University of Technology, Mail H74 PO Box 218, Hawthorn, VIC 3122, Australia. [12] These authors contributed equally: Harrisson D. A. Santos, Irene Zabala Gutiérrez, Yingli Shen, José Lifante and Erving Ximendes. ✉email: bjrubio@ucm.es; daniel.jaque@uam.es

Semiconductor nanocrystals emerged almost two decades ago as novel fluorescent contrast agents capable of overcoming the intrinsic limitations of traditional fluorescent biomarkers (dyes and fluorescent proteins)[1,2]. The demonstration of the ability of these nanocrystals to generate high-contrast images at the cellular level was a major milestone in biophotonics and promoted further research into these materials. Optimizing their chemical stability, biocompatibility, and brightness was the major focus of the subsequent investigation[3,4]. That resulted in the development of core/shell architectures, which constituted the second generation of semiconductor nanocrystals[5,6]. These produced unparalleled results in cell imaging[7], but failed to enable deep-tissue in vivo imaging as biological tissues are efficient absorbers and scatterers of visible light. This limitation was overcome by the development of a third generation of semiconductor nanocrystals, emitting in the second near-infrared window (NIR-II, 1000–1700 nm)[8]. In this spectral range, the absorption and scattering coefficients of biological tissues are reduced to a minimum, allowing high-contrast, high-resolution in vivo imaging at large (>1 cm) tissue depths[9]. NIR-II fluorescent materials have changed the game in preclinical imaging[10,11], enabling high-resolution anatomical imaging[12,13], tumor detection[14,15], biosensing[16], brain vasculature mapping[17,18], image-guided genome editing[19] and surgery[20,21], and dynamic tracking of metabolic processes[22]. NIR-II-emitting Ag$_2$S nanocrystals do not contain highly toxic heavy metal ions, unlike other nanoprobes operating in this spectral range, minimizing biocompatibility concerns and making them one of the most promising systems among all currently reported NIR-II fluorophores[23,24]. Their properties have been exploited for multiple applications, including subcutaneous and transcranial thermometry[25], tumor theranostics via photoacoustic imaging and photothermal therapy[26], early tumor diagnosis[27] and dynamic imaging of the heart[28] and the cardiovascular system[29–31].

To enhance the potential of Ag$_2$S dots as NIR-II contrast agents, their properties need substantial improving to enable deep-tissue imaging at irradiation intensities well below the established safety threshold. Currently available Ag$_2$S dots are limited by their low fluorescence brightness, which is a result of their low quantum yield (QY < 1%) and short fluorescence lifetime (<100 ns)[32]. These features have been attributed to the presence of surface and structural defects and to the dot–solvent interactions that favor electronic deexcitation via non-radiative pathways[33]. Furthermore, the high redox potential of silver ions and the high temperatures required for Ag$_2$S dots synthesis lead to the simultaneous formation of metallic silver nanoparticles (NPs) that could also reduce the overall brightness due to plasmon coupling events[34]. Currently available chemical synthesis routes have failed to avoid these non-radiative channels in Ag$_2$S dots. Therefore, developing alternative approaches is essential to overcome this barrier.

In this work, we present a novel methodology that permits an 80-fold increment in the QY of Ag$_2$S dots. The process is based on the irradiation of chloroform-dispersed Ag$_2$S dots with femtosecond laser pulses, which leads to the formation of a protective AgCl shell. This reduces the surface traps while minimizing dot-to-medium energy transfer via non-radiative events. The brightness of the NPs generated in this process is superior to that of all other currently available NIR-II contrast agents. Their improved performance enables in vivo whole body imaging, blood vessel visualization, and biodistribution tracking at ultra-low excitation intensities.

## Results

### Preparation and characterization of Ag$_2$S superdots. We fabricated Ag$_2$S superdots by irradiating a dispersion of chemically

synthesized Ag/Ag$_2$S heterodimers (hereafter Ag$_2$S dots) in CHCl$_3$ with femtosecond laser pulses. The chemical synthesis route we used (described in Methods) yields silver NPs as a side product (see Supplementary Fig. 1). The presence of these NPs is due to the high redox potential of silver ions at the reaction temperature[35]. To eliminate them, a 1 mg mL$^{-1}$ CHCl$_3$ dispersion of Ag$_2$S dots was irradiated with a Ti:Sapphire femtosecond laser operating at 808 nm (see details in Methods). Laser pulses trigger the transformation of Ag NPs into AgCl particles, which are colloidally unstable and easy to remove (see Supplementary Fig. 2).

Fig. 1 shows the morphological and compositional analysis of the Ag$_2$S dots before and after ultrafast laser irradiation. Fig. 1a depicts a high-angle annular dark-field scanning transmission electron microscopy (HAADF-STEM) image of the as-synthesized Ag$_2$S dots. They present an elliptical morphology with an average size of 9.5 ± 1.0 nm (see Supplementary Fig. 3). The NPs present an eccentrically located, electron-dense core embedded in a matrix of lower electron density. The high-resolution STEM of a typical particle reveals a crystalline structure (Fig. 1b), with lattice fringes d$_{-104}$ = 2.37 Å and, in the electron-dense core, d$_{111}$ = 2.30 Å. These well match monoclinic Ag$_2$S (JCPDS No. 14-0072) and cubic Ag (JCPDS 04-0783), respectively. The less electron-dense area in Fig. 1b is postulated to be Ag$_2$S, as Ag$_2$S has a lower density (7.2 g cm$^{-3}$) than Ag (10.505 g cm$^{-3}$). Energy-dispersive X-ray spectroscopy (EDS) analysis, shown in Fig. 1c–e, supports this conclusion. While the NP core is rich in Ag, S is mainly located in the outer part of the NPs. This anisotropic distribution of elements is depicted in the net X-ray intensity profiles shown in Fig. 1g as obtained from the magnified STEM micrograph shown in Fig. 1f. In Fig. 1g, we can observe that the maximum of the X-ray intensity assigned Ag and S atoms arises, respectively, from the electron-dense core and the outer area. The elemental analysis of a Ag$_2$S dot shows a Ag:S ratio of 74:26. All these results support the presence of a metallic Ag core within the Ag$_2$S matrix.

After ultrafast illumination with 50 fs pulses for 90 min (9 W cm$^{-2}$), the average diameter of the NPs increased from 9.5 ± 1.0 to 12.3 ± 1.0 nm, while the size of the metallic Ag core remained constant at around 5.1 nm (see Fig. 1h and Supplementary Fig. 3). High-resolution STEM of a representative Ag$_2$S dot after ultrafast laser irradiation (Fig. 1i) reveals, again, two well-differentiated crystalline regions. An electron-dense core, with lattice fringe of d$_{111}$ = 2.30 Å that agrees with metallic Ag is embedded in a matrix with a lattice fringe of d$_{110}$ = 2.50 Å, corresponding to Ag$_2$S. The presence of a 1-nm-thick shell around the NPs can also be observed in this figure. Based on the analysis of the corresponding crystal lattices, the conjunction interfaces between the metallic and the semiconductor part consist of the (110) plane of Ag$_2$S and (111) of Ag, with a lattice mismatch of 8%. EDS images (Fig. 1j–m) revealed that the resulting NPs are composed of Ag and S with an identical distribution as that observed in the as-synthesized NPs. Further, EDS analysis indicated the presence of Cl atoms distributed throughout the laser-irradiated Ag$_2$S dots (Fig. 1l). This stems from the formation of a AgCl shell. The presence of this shell is also evidenced in the net X-ray intensity profiles shown in Fig. 1o, obtained from the magnified STEM micrograph shown in Fig. 1n. The elemental analysis of a laser-treated Ag$_2$S dot shows a Ag:S:Cl ratio of 69:24:7. Again, the excess of Ag would indicate the conservation of the silver-rich core characteristic of the Ag$_2$S dots. Further chemical and structural characterization of the samples, including X-ray diffraction, X-ray absorption near edge structure (XANES), EDS, and HAADF-STEM data of both Ag$_2$S dots and superdots is provided in Supplementary Figs. 4 and 5. These additional data support the conclusions extracted from Fig. 1 that the

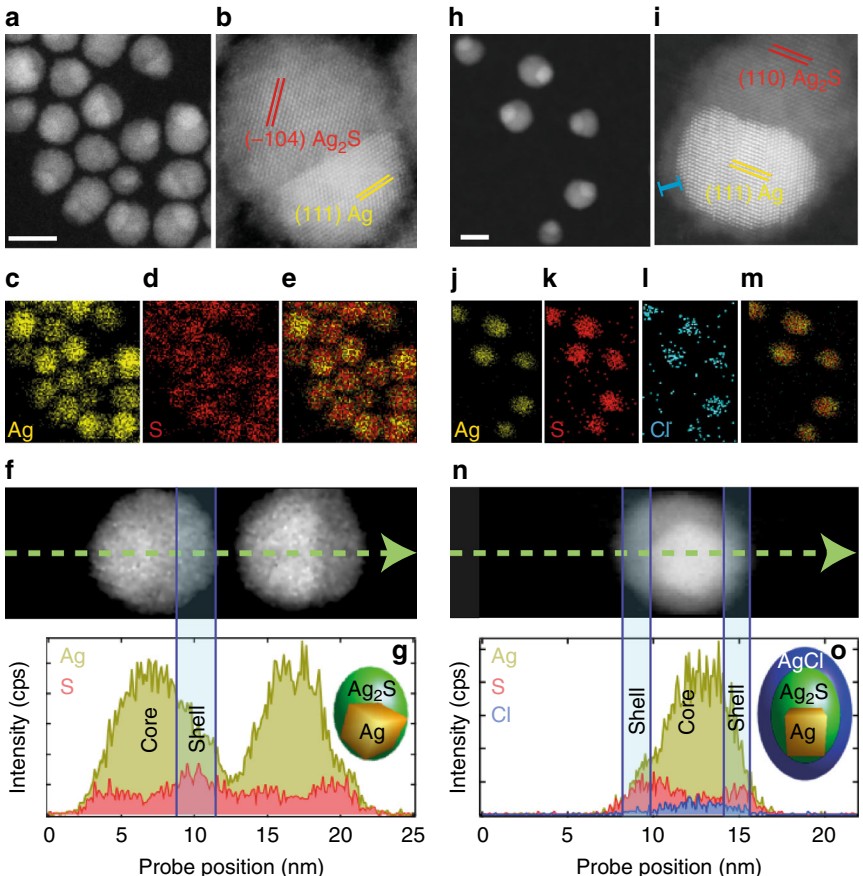

**Fig. 1 Tuning structural and chemical properties through ultrafast laser irradiation. a** HAADF-STEM micrograph of the as-synthesized $Ag_2S$ dots. Scale bar is 10 nm. **b** High-resolution STEM of as-synthetized dots showing lattice fringes $d_{111}$ and $d_{-104}$ of cubic Ag and monoclinic $Ag_2S$, respectively. 2D EDS mapping of the spatial distribution of Ag (**c**), S (**d**), and Ag+S (**e**). **f** Magnified STEM micrograph of two $Ag_2S$ dots. **g** Net X-ray intensity profiles extracted from the green arrow marked in **f**. Note how the Ag/S ratio increases at the edges of the dots, which coincides with the electro-dense area. The inset shows a model $Ag_2S$ dot. **h** HAADF-STEM micrograph of $Ag_2S$ dots after ultrafast laser irradiation with 50 fs laser pulses for 90 min at a power density of 9 W cm$^{-2}$. Scale bar is 10 nm. **i** High-resolution STEM of an ultrafast laser-irradiated $Ag_2S$ dot showing lattice fringes $d_{111}$ and $d_{110}$ of cubic Ag and monoclinic $Ag_2S$, respectively. Note the presence of a shell marked with a blue bracket around the nanoparticle. 2D EDS mapping of the spatial distribution of Ag (**j**), S (**k**), Cl (**l**), and Ag+S + Cl (**m**) of an ultrafast laser-irradiated $Ag_2S$ dot. **n** Magnified STEM micrograph of an ultrafast laser-irradiated $Ag_2S$ dot. **o** Net X-ray intensity profiles extracted from the green arrow marked in image (**n**), where we can observe the presence of Ag, S, and Cl. The inset in **o** shows a schematic representation of a single $Ag_2S$ dot after ultrafast laser irradiation.

as-synthesized dots are composed of two phases, a monoclinic $Ag_2S$ phase and a cubic Ag phase. When these NPs are irradiated with an ultrafast laser, their chemical composition and structure change. The resulting NPs exhibit a new AgCl phase forming a thin outer shell, as observed in Fig. 1i.

**Optical transformation of $Ag_2S$ dots into superdots**. The structural changes induced in the $Ag_2S$ dots by ultrafast laser pulses, summarized in Fig. 1, are accompanied by a dramatic change in their optical properties. This change is visible to the naked eye, as seen in Fig. 2a, which contains the optical images of a colloidal dispersion of as $Ag_2S$ dots in $CHCl_3$ before and after ultrafast laser irradiation (50 fs, 90 min, 9 W cm$^{-2}$). The dispersion changes color as a result of the ultrafast laser irradiation, becoming progressively more transparent (see Supplementary Fig. 6). Ultrafast laser irradiation reduces the absorbance of the sample in the 400–800 nm range by 35%, as seen in Fig. 2b. This reduction occurs progressively during ultrafast laser irradiation, as shown in Supplementary Fig. 6. The absorption spectrum of the as-synthesized $Ag_2S$ dots presents a shoulder around 400 nm, which corresponds to the plasmonic band of sub-100-nm Ag

NPs[36]. This shoulder is no longer present after ultrafast laser irradiation. Besides modulating the optical absorption, ultrafast laser irradiation enhances the fluorescence brightness of the $Ag_2S$ dots, as shown in the NIR-II fluorescence images in Fig. 2c. These correspond to a colloidal dispersion of $Ag_2S$ dots before and after ultrafast laser irradiation (50 fs, 90 min, 9 W cm$^{-2}$). The emission of the as-synthesized $Ag_2S$ dots is barely detectable with our NIR-II imaging system, in contrast to the high signal observed after the irradiation process. This enhancement in the NIR-II brightness occurs progressively during ultrafast irradiation, as seen in Fig. 2d, which shows the time evolution of the NIR-II emission of $Ag_2S$ dots during the process. The shape of the emission spectrum, as seen in the inset in Fig. 2d, remains virtually unchanged. This result indicates that the chemical nature of the fluorophore does not change upon irradiation. Fig. 2e shows the fluorescence QY of different colloidal dispersions of $Ag_2S$ dots in $CHCl_3$ after different durations of ultrafast laser irradiation (50 fs, 9 W cm$^{-2}$). Supplementary Fig. 7 includes some representative excitation and emission spectra used for QY calculations. The QY increases monotonously with the irradiation time for treatment durations shorter than 50 min, while longer irradiations do not lead to any further increase. The QY of as-synthesized $Ag_2S$ dots increases

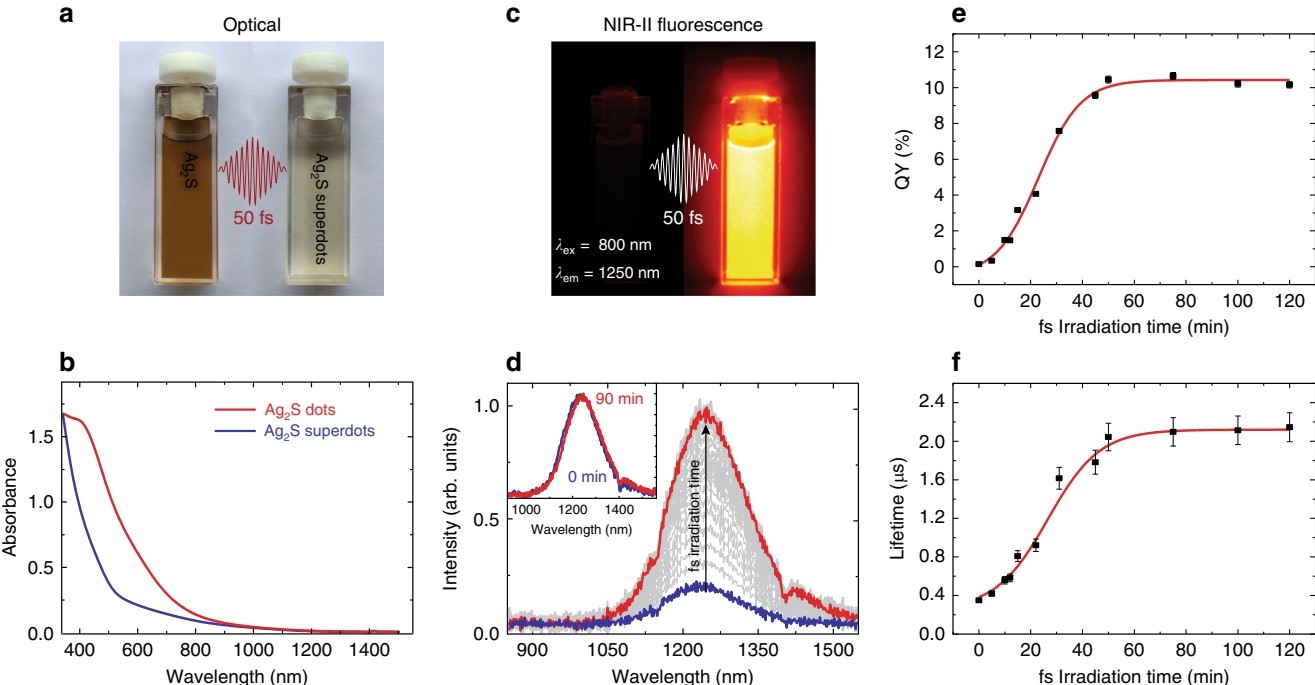

**Fig. 2 Optical transformation of Ag₂S dots into superdots. a** Optical images, **b** UV-VIS-NIR extinction spectra, and **c** NIR-II fluorescence images of a colloidal dispersion of $Ag_2S$ dots in $CHCl_3$ before and after ultrafast laser irradiation (50 fs, 90 min, 9 W cm$^{-2}$). For the acquisition of NIR-II fluorescence images, the dispersions were optically excited with an 808 nm continuous wave laser diode (100 mW cm$^{-2}$). **d** NIR-II emission spectra generated by a colloidal suspension of dots in $CHCl_3$ during a 90-min-long ultrafast laser irradiation. The inset shows the normalized emission spectra of the sample before and after the laser irradiation. **e** Fluorescence quantum yield and **f** fluorescence lifetime of $Ag_2S$ dots in $CHCl_3$ after being subjected to ultrafast laser irradiations processes of different durations. The error bars in **e** were calculated taking into account the equipment uncertainty that was determined from the statistical analysis of five measurements. In **f**, each measurement was repeated five times and the error bars are the standard error of the mean of each series of measurements. Repetitive measurements to calculate both mean and error were performed on the same sample. The small relative uncertainties in QY (2%) make the error bars too small to be visualized in **e**. In **f**, the error bars correspond to the standard error of the mean after measuring and analysing 10 decay curves for each sample. In **e** and **f**, the squares correspond to the experimental data and the red lines are guides for the eyes.

from 0.13 to 10.7% in a 50-min-long irradiation with 50 fs pulses. This constitutes an 80-fold enhancement, validating our referring to the laser-irradiated $Ag_2S$ dots as superdots. This QY enhancement is accompanied by a substantial increase in the fluorescence lifetime from 200 ns up to 2.1 μs, as shown in Fig. 2f (see Supplementary Fig. 6 for the decay curves). This lifetime increase follows the same trend with the irradiation time observed for the QY in Fig. 2e and is independent of the dot concentration in the 0.1–1 mg mL$^{-1}$ range, as shown in Supplementary Fig. 8. These results rule out any possible contribution of fluorescence self-absorption to our lifetime measurements.

**Mechanism of dot-to-superdot transformation**. The dot-to-superdot transformation induced by ultrafast laser irradiation depends critically on multiple experimental variables. Fig. 3a shows the time evolution of the NIR-II fluorescence intensity generated by a colloidal dispersion of dots in $CHCl_3$ during ultrafast laser irradiation at different irradiation power densities for irradiation times up to 100 min. The curves in Fig. 3a were obtained for constant pulse duration (50 fs), repetition rate (1 kHz), and average power (0.6 W). For irradiation power densities below 3 W cm$^{-2}$, which seems to be a threshold value, there is no apparent improvement in the luminescence properties. For irradiation power densities between 3 and 8 W cm$^{-2}$, the NIR-II fluorescence intensity increases with the irradiation time. In this range of power densities, the slope of the intensity vs. time curve ($\eta = dI/dt$) strongly depends on the irradiation pulse energy ($E_p$). In fact, $\eta$ is proportional to $E_p^2$, which suggests that the dot-to-superdot

transformation is triggered by a two-photon absorption process (see Supplementary Note 1 and Supplementary Fig. 9). For irradiation power densities between 8 and 9 W cm$^{-2}$, the NIR-II fluorescence intensity increases with irradiation time until a stable value is reached. Finally, irradiation at or above 10 W cm$^{-2}$ leads to a decrease in the emitted intensity for long irradiation times that may be attributed to sample degradation. This effect is highlighted for irradiation power densities close to 100 W cm$^{-2}$. Under these conditions, an initial increase in the emission intensity is followed by an abrupt reduction, rendering a nonluminescent and completely transparent dispersion. Plotting the fluorescence intensity of the irradiated solution as a function of the irradiation power density evidences the existence of an optimum irradiation power density close to 9 W cm$^{-2}$ (see Supplementary Fig. 10).

Fig. 3b demonstrates that ultrafast laser-induced dot-to-superdot transformation only occurs when the as-synthesized $Ag_2S$ dots are dispersed in $CHCl_3$. When they are dispersed in water, toluene, or hexane, pulsed irradiation leads to a decrease in their NIR-II fluorescence that can be attributed to laser-induced thermal loading (see Supplementary Note 2 and Supplementary Fig. 11). The presence of silver is also critical for the dot-to-superdot transformation. We corroborated this fact by analysing the effect of ultrafast laser irradiation in a $Ag_2S$ dot dispersion free of Ag NPs (see Methods and Supplementary Fig. 12 for details). Ultrafast laser irradiation of these dots does not improve their fluorescence, as evidenced in Fig. 3c. Thus, ultrafast laser-driven dot-to-superdot transformation requires not only the presence of $CHCl_3$ but also of Ag NPs in the dispersion.

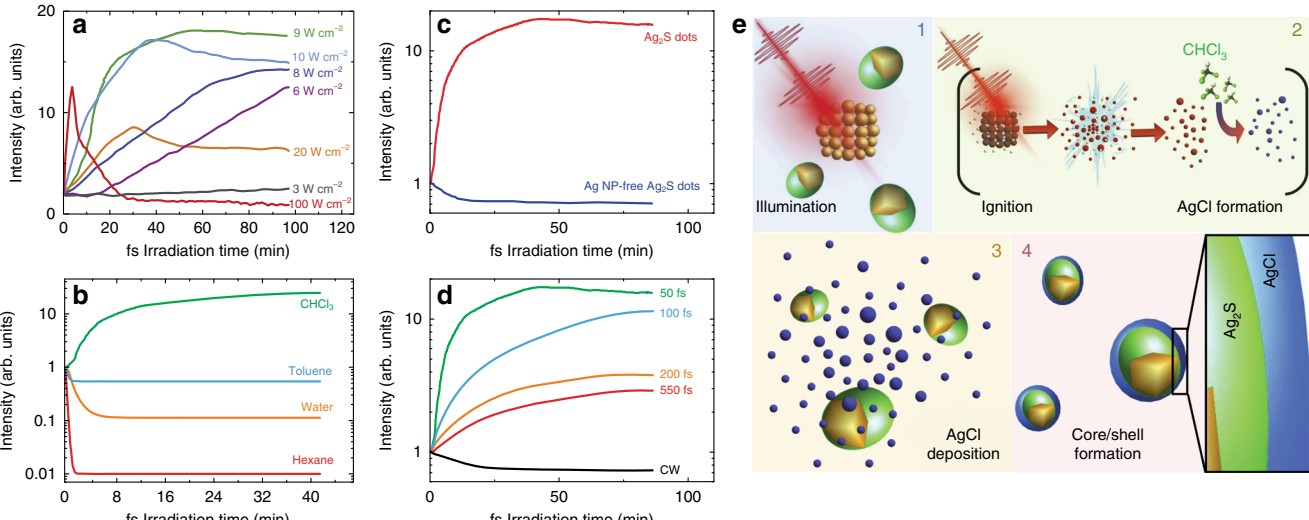

**Fig. 3 Required conditions and mechanisms of laser-induced dot-to-superdot transformation. a** Time evolution of the NIR-II fluorescence generated by colloidal dispersions of $Ag_2S$ dots in $CHCl_3$ under irradiation at different power densities. In all cases, pulse width and repetition rate were set to 50 fs and 1 kHz, respectively. **b** Time evolution of the NIR-II fluorescence generated by $Ag_2S$ dots dispersed in different solvents under laser irradiation with 50 fs, 808 nm laser pulses. **c** Time evolution of the NIR-II fluorescence generated by $Ag_2S$ dots under laser irradiation with 50 fs, 808 nm laser pulses in presence and absence of Ag NPs. **d** Time evolution of the NIR-II fluorescence intensity generated by $Ag_2S$ dots in $CHCl_3$ as obtained for different pulse durations. For all cases in **b**, **c**, and **d**, the ultrafast laser power density was set to 9 W cm$^{-2}$. Different samples were used in each condition used in **a**, **b**, **c** and **d**. **e** Schematic representation of the physicochemical mechanisms underlying the ultrafast laser-induced dot-to-superdot transformation. Upon illumination with ultrafast infrared laser pulses, multiphoton excitation of Ag NPs leads to their Coulomb explosion (step 1). The $Ag^{+z}$ generated in this process react with $CHCl_3$ molecules forming AgCl (step 2), which in turn reacts with the surface of the $Ag_2S$ dots forming a protective layer (steps 3 and 4).

Moreover, the dynamics of the dot-to-superdot transformation depends critically on the duration of the laser pulses. Fig. 3d shows the time evolution of the NIR-II intensity emitted by a dispersion of $Ag_2S$ dots in $CHCl_3$ during irradiation with laser pulses of different durations and identical power (0.6 W) and power density (9 W cm$^{-2}$). The efficiency of the dot-to-superdot transformation, estimated from the magnitude of the enhancement in the NIR-II emission, decreases for longer pulse durations. Irradiation with a continuous wave laser does not improve the NIR-II fluorescence brightness, indicating that no dot-to-superdot transformation occurs.

The experimental results shown in Figs. 1–3 allow us to provide a plausible explanation for the dot-to-superdot transformation. This is schematically shown in Fig. 3e. Due to the synthesis route used here, the as-prepared dispersions of $Ag_2S$ dots also contain Ag NPs. The plasmon resonance of these Ag particles is responsible for the extinction peak observed at around 400 nm (see Fig. 2b). When excited with 808 nm ultrafast pulses, two-photon absorption by Ag NPs induces a high free electron density that results in the Coulomb explosion of the Ag NPs (step 1 in Fig. 3e)[37]. The key role of multiphoton excitation is supported by the requirement of ultrafast laser pulses that ensure high photon densities. These allow for sequential absorption through virtual states of Ag NPs[38]. The participation of two 808 nm photons in this process is further supported by the experimental data in Supplementary Fig. 9. The Coulomb explosion of Ag NPs leads to an increment in the concentration of highly reactive $Ag^+$ in the solution, which react with $CHCl_3$ yielding silver chloride (AgCl, step 2 in Fig. 3e). The laser-generated AgCl molecules interact with the surface of the $Ag_2S$ dots, where a protective a AgCl shell is formed (steps 3 and 4 in Fig. 3e). Therefore, after ultrafast laser irradiation, the low-bandgap semiconductor $Ag_2S$ (0.9 eV) is coated with an inorganic shell of higher-bandgap AgCl (5.13 eV) several monolayers thick, as seen in Fig. 1i[39]. This protective shell strongly reduces the non-radiative transitions that involve the vibronic activation of solvent $CHCl_3$ molecules and prevents the formation of shallow or deep midgap states as surface traps that would provide non-radiative deexcitation pathways. The reduction in the non-radiative decay probabilities caused by this protective shell simultaneously explains the increment in the NIR-II QY and the fluorescence lifetime. The decrease in the non-radiative decay probability is also evidenced by a reduction in the light-to-heat conversion efficiency when the dot-to-superdot transformation takes place (see Supplementary Fig. 13 and Supplementary Note 3). Both dots and superdots can be damaged under excessive irradiation doses, explaining why at high irradiation power densities the initial enhancement in NIR-II luminescence is followed by a nonreversible quenching (see Fig. 3a). An arising question is why the multiphoton excitation and subsequent Coulomb explosion only occur in the Ag NPs and not in the Ag cores within the $Ag_2S$ dots. A plausible explanation is that Ag cores have no plasmonic resonance that can be excited by 808 nm fs laser pulses due to being surrounded by $Ag_2S$. The very different dielectric constants/refractive indices of $CHCl_3$ and $Ag_2S$ could be behind this effect, since the spectral location of the plasmon resonance of Ag NPs strongly depends on the dielectric constant/refractive index of the surrounding medium[40]. The absence of multiphoton excitation of the Ag core would prevent its Coulomb explosion, required for the formation of the AgCl shell[41].

**In vivo NIR-II imaging with $Ag_2S$ superdots**. To test the potential application of $Ag_2S$ superdots for in vivo NIR-II imaging, we transferred them from their original solvent ($CHCl_3$) to phosphate buffer saline (PBS) by means of a ligand exchange procedure described in Methods. The adhesion of hydrophilic and bifunctional HS-PEG-COOH ligands on the surface of $Ag_2S$ superdots makes them stable in PBS with no signs of precipitation for at least 12 months. The average hydrodynamic diameter and Z-potential after ligand exchange are 22 nm, and −25 mV, respectively. PEG coating and dispersion in PBS does not significantly reduce the fluorescence

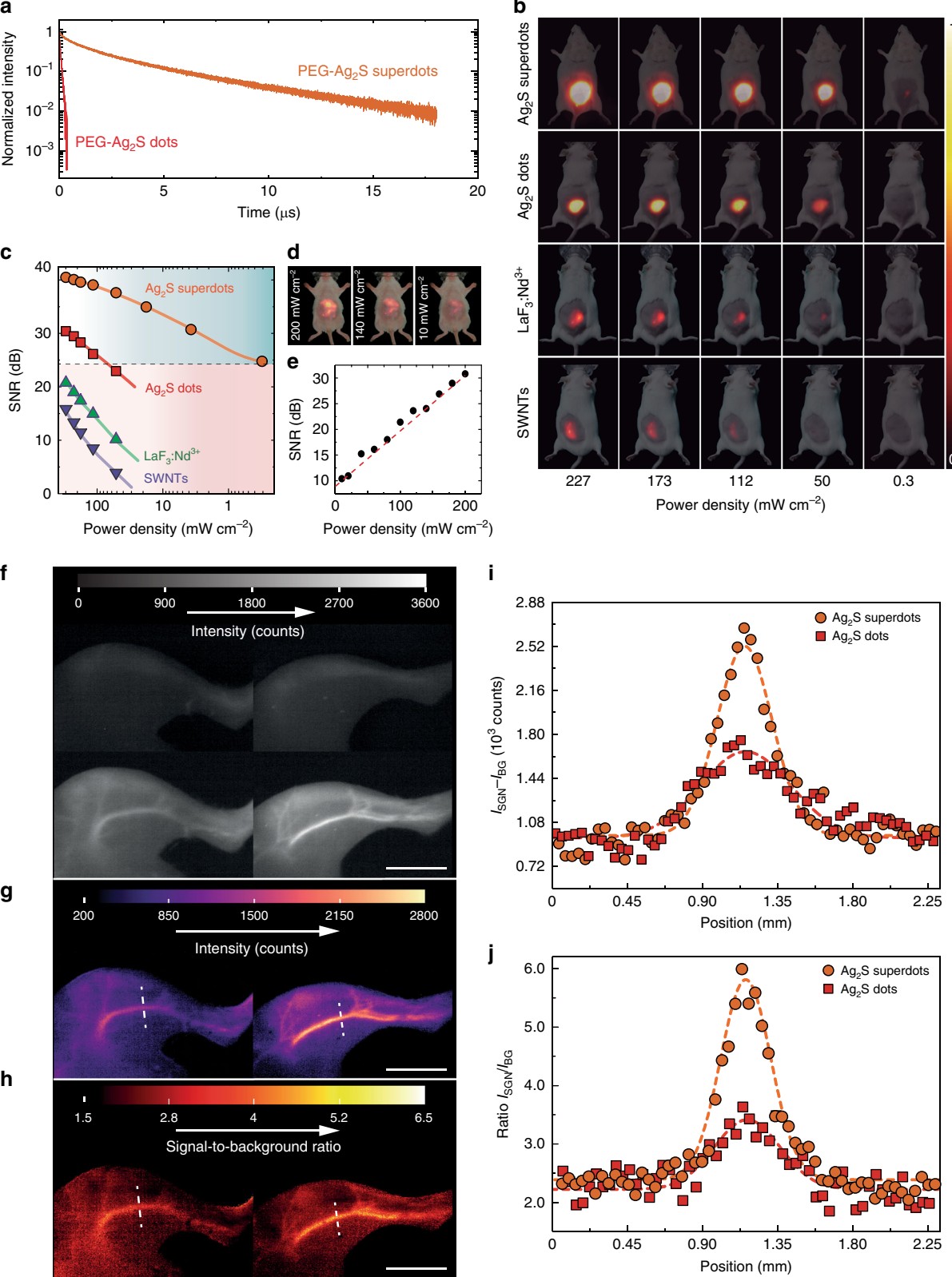

decay time of Ag$_2$S superdots, which is close to 2 μs (see Fig. 4a). The fact that Ag$_2$S superdots show the same decay time in CHCl$_3$ and PBS reveals the effectiveness of the AgCl protective shell against the appearance of multiphonon relaxation events favored by the vibration modes of water molecules. This, in turn, results in identical QY values (≈10%) in both solvents. Fig. 4a also includes, for the sake of comparison, the

fluorescence decay curve obtained from commercially available (Sinano Corp. China) PEG-coated Ag$_2$S dots dispersed in PBS. These show a fluorescence lifetime of 53 ns, about 40 times shorter than that of PBS-dispersed PEG-coated Ag$_2$S superdots. Such a long fluorescence lifetime and high QY suggests the potential application of PEGylated Ag$_2$S superdots for high-contrast, low-dose NIR-II in vivo imaging.

**Fig. 4 In vivo brightness of Ag$_2$S superdots: a comparison with their competitors. a** NIR-II fluorescence decay curves of PEG-coated Ag$_2$S dots (provided by Sinano Corp., China) and Ag$_2$S superdots, both dispersed in PBS. **b** NIR-II fluorescence images of a group of four mice subcutaneously injected with colloidal aqueous dispersions containing Ag$_2$S superdots, commercial Ag$_2$S dots, SWNTs, and LaF$_3$:Nd NPs. The same NP volume (100 μL) and concentration (1.5 mg mL$^{-1}$) were injected in all cases. The different images for same optical probe correspond to different 808 nm illumination power densities. **c** Signal-to-noise ratio (SNR) as a function of power density quantified from the analysis of images in **b** for the four NIR-II nanoprobes evaluated. **d** Fluorescence images of Ag$_2$S superdots accumulated in the liver obtained at three different 808 nm excitation power densities. **e** Power density dependence of SNR calculated from in vivo NIR-II fluorescence images at different excitation power densities. **f** NIR-II fluorescence images of the left hind limbs of two mice immediately before (top) and 15 s after (bottom) of an intravenous injection of Ag$_2$S dots (left) or superdots (right). **g** Net intensity images obtained from subtracting the background images (top row in **f**) from the signal images (bottom row in **f**). **h** Signal-to-background images obtained by dividing the signal intensity images (bottom row in **f**) by the background images (top row in **f**). Scale bars in **f**, **g**, and **h** are 2 mm. **i** Net intensity and **j** signal-to-background ratio obtained along a line profile across the saphenous artery (indicated as dashed white lines in **g** and **h**).

To evaluate the performance of Ag$_2$S superdots in in vivo imaging, we benchmarked them against other well-established NIR-II fluorescent probes: commercial Ag$_2$S dots, neodymium-doped nanocrystals (LaF$_3$:Nd$^{3+}$), and single-walled carbon nanotubes (SWNTs)[42–44]. FDA-approved indocyanine green was not included in this comparison as preliminary experiments revealed the poor spectral overlap between its emission tail and the spectral response of our NIR-II imaging system (see Supplementary Figs. 14 and 15)[45,46]. For our comparative study, we used four different mice that were subcutaneously injected with 100 μL of a PBS solution containing, in each case, Ag$_2$S superdots, Ag$_2$S dots, LaF$_3$:Nd$^{3+}$, and SWNTs. The NP concentration was set in all cases to 1.5 mg mL$^{-1}$, corresponding to a total injected NP dose of ≈5 mg kg$^{-1}$. Fig. 4b shows the in vivo NIR-II fluorescence images obtained in each case for 808 nm illumination power densities ranging from 227 down to 0.3 mW cm$^{-2}$. These results indicate that Ag$_2$S superdots outperform all other NIR-II fluorescent probes evaluated here from a brightness standpoint (see Supplementary Table 1). We considered the possibility of experimental error in these results due to individual variations in the thickness of the mouse skin. However, this would only account for 20% of the total variation in the transmitted fluorescence (±80 μm over a 400 μm layer of skin, see Supplementary Note 4). These variations are negligible compared with the dramatic difference in the emission intensity observed experimentally between the Ag$_2$S superdots and the rest of the evaluated fluorescent probes. The superior brightness of the superdots enables the acquisition of fluorescence images with sufficient contrast even at the ultra-low irradiation power density of 0.3 mW cm$^{-2}$, where no fluorescence from any of the other three tested NIR-II probes could be registered. This not only allows in vivo imaging with cost-effective excitation sources but also ensures a negligible thermal loading during image acquisition (see Supplementary Fig. 16)[38]. The superior contrast achieved imaging with Ag$_2$S superdots is quantified in Fig. 4c, which shows the power density dependence of the signal-to-noise ratio (SNR) calculated from the fluorescence images in Fig. 4b. From Fig. 4c, we conclude that the brightness improvement achieved during ultrafast laser irradiation reduces the minimum illumination density that can be used for in vivo imaging by almost two orders of magnitude. This enhancement is also evident when comparing the illumination conditions used in this work to those reported previously for Ag$_2$S dots, SWNTs, and lanthanide-doped nanocrystals (see Supplementary Table 2).

The data in Fig. 4c correspond to the analysis of in vivo images of subcutaneous injections. To provide a realistic figure of the minimum power density required for in vivo imaging, we performed similar experiments for intravenously injected Ag$_2$S superdots. Injection volume and concentration were set to 100 μL and 0.15 mg mL$^{-1}$, respectively. Fig. 4d shows the in vivo fluorescence images for three different excitation power densities obtained 30 min after intravenous injection of a dispersion of Ag$_2$S

superdots. At this time point, the superdots had accumulated mostly in the liver and the spleen. Fig. 4e shows the linear dependence of the SNR with the power density, indicating that even at a very low excitation intensity (10 mW cm$^{-2}$) the SNR is above 10 dB. This is a major improvement with respect to the intensities required for in vivo imaging with Ag$_2$S dots (see Supplementary Table 2), which are typically above 100 mW cm$^{-2}$. The high brightness of Ag$_2$S superdots also allows imaging at large depths (>1 cm) into tissues (see Supplementary Note 5 and Supplementary Fig. 17).

Ag$_2$S superdots enable improved in vivo visualization of blood vessels, as shown in Fig. 4f–j. The images in Fig. 4f–h correspond to the left hind limbs of two mice after intravenous injection of either Ag$_2$S dots or superdots (see Methods). Injection volume and concentration were set to 100 μL and 0.15 mg mL$^{-1}$, respectively. Fig. 4f shows the magnified luminescence images of the limb immediately before (top) and 15 s after injection (bottom). Although the preinjection background (caused by tissue autofluorescence) was similar in both cases, the image contrast after injection is substantially higher after injection of superdots (right) than in the case of dots (left). We quantified the improvement in image contrast by calculating the signal-to-background ratio for both sets of images. This was done by subtracting the background from the overall emission to obtain net signal images (Fig. 4g), which were also divided by the background to calculate the signal-to-background ratios (Fig. 4h). Both the net signal ($I_{SG} - I_{BG}$, Fig. 4i) and the signal-to-background ($I_{SG}/I_{BG}$, Fig. 4j) line profiles along the saphenous artery (dashed lines in Fig. 4g, h), indicated a superior performance of the Ag$_2$S superdots as compared with dots. The pixel profiles in Fig. 4i, j indicate that Ag$_2$S superdots improve the net signal by 60% and the signal-to-background ratio by 90% when compared with conventional Ag$_2$S dots. Further discussion on the NIR-II images, focusing on the relative quality of the ones presented here to those previously reported by other groups[15,16,18,29,31,47], is included in Supplementary Note 6, Supplementary Fig. 18 and Supplementary Table 3.

Ag$_2$S superdots also outperform conventional Ag$_2$S dots at in vivo video rate NIR-II fluorescence imaging. Fig. 5a shows the NIR-II images obtained at different times after intravenous administration of PEG-coated Ag$_2$S superdots (100 μL, 0.15 mg mL$^{-1}$). The total dose (15 μg, ~0.5 mg kg$^{-1}$) is more than one order of magnitude smaller than the administered dose (6.6 mg kg$^{-1}$) reported for NIR-II video rate recording with Ag$_2$S dots[29]. The use of low administration doses for in vivo imaging is beneficial for different reasons. First, to develop a cost-effective probe for NIR-II in vivo imaging, the amount of material required to achieve high contrast in vivo images should be reduced. Low administration doses minimize potential toxicity of the probe. Further, Ag$_2$S superdots show some photothermal effect that requires low administration doses and illumination intensities to prevent undesirable heating during in vivo imaging. For the conditions used in this work,

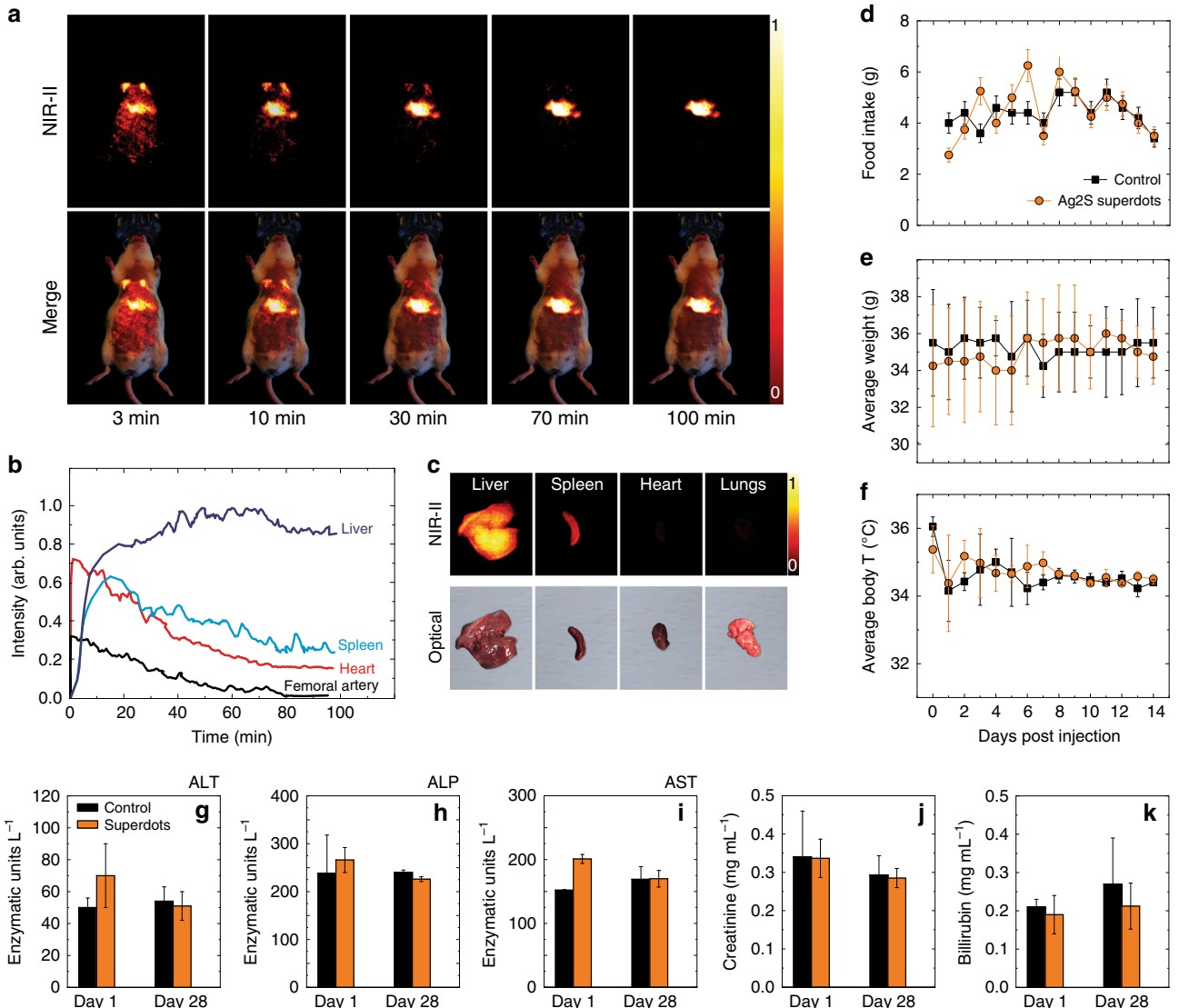

**Fig. 5 In vivo time-resolved imaging with Ag₂S superdots. a** NIR-II fluorescence images obtained at different times after intravenous injection of Ag₂S superdots dispersed in PBS (100 μL, 0.15 mg mL⁻¹). **b** Time evolution of the fluorescence intensity at the liver, spleen, heart, and femoral artery after intravenous injection of Ag₂S superdots. **c** NIR-II ex vivo fluorescence and optical images of the liver, spleen, heart, and lungs obtained from a mouse euthanized 100 min after intravenous injection of Ag₂S superdots. Time evolution of daily food intake (**d**), weight (**e**), and body temperature (**f**) of four CD1 mice after intravenous injection of 300 μL of a dispersion of Ag₂S superdots in PBS (0.5 mg mL⁻¹, total dose of 150 μg). The results obtained from a control group intravenously injected with 300 μL of PBS are included for comparison. Serum concentration (**g**) of hepatic enzymes (**g, h** and **i**) for mice intravenously injected with Ag₂S superdots and control mice as obtained 1 and 28 days after injection. Serum concentration of creatinine (**j**) and bilirubin (**k**) corresponding to mice subjected to an intravenous injection of Ag₂S superdots and control mice as obtained 1 and 28 days after injection. ($n = 3$ for each group). The error bars correspond to the standard error of the mean (±SEM).

there is negligible thermal loading due to the photothermal conversion of Ag₂S superdots (see Supplementary Fig. 19). The images shown in Fig. 5a were obtained under 808 nm excitation at 10 mW cm⁻², which is one order of magnitude lower than the illumination power density reported for NIR-II video recording using SWNTs (140 mW cm⁻²)[43]. The illumination power density here used is also much lower than the safety threshold at this wavelength (329 mW cm⁻²) established by the International Commission on Non-ionizing Radiation Protection (ANSI Z136.1-2000)[48]. NIR-II video recording enabled us to track the in vivo biodistribution of our Ag₂S superdots. The time evolution of the NIR-II fluorescence intensity generated by the Ag₂S superdots at the liver, spleen, heart, and femoral artery after intravenous injection is shown in Fig. 5b. During the first 3 min after injection, the PEG-coated superdots were circulating and

gathering in the liver, spleen, and heart. The fluorescence signal observed at the femoral artery is assigned to the presence of Ag₂S superdots in the bloodstream. Fitting this curve to a first-order exponential, we estimate a blood half-life close to 20 min for the Ag₂S superdots. As seen in Fig. 5a, b, most of the superdots have been uptaken by the liver and spleen after 30 min. This is supported by ex vivo NIR-II images taken 100 min after injection (Fig. 5c). The accumulation of NPs in the liver and spleen has been widely reported and is related to filtration mechanisms promoted by the reticuloendothelial system[49].

To evaluate the in vivo biocompatibility of our Ag₂S superdots, we analysed the time evolution of weight, food intake, and body temperature of four CD1 mice for 2 weeks after intravenous administration of Ag₂S superdots dispersed in PBS. The injected dose (150 μg, equivalent to 5 mg kg⁻¹) is one order of magnitude

above that used for NIR-II in vivo video recording. Four additional mice were injected with 300 μL of PBS and used as a control. Experimental data included in Fig. 5d–f reveal that, even at such relatively high administration doses, there are no relevant differences between both groups. These findings are consistent with the reported low toxicity of conventional $Ag_2S$ dots and with the already demonstrated ability of AgCl coatings to minimize NP cytotoxicity[50,51]. To further test the long-term biocompatibility of $Ag_2S$ superdots, we performed a 28-day subchronic toxicological experiment and histological analyses, as detailed in the Supplementary Methods. Supplementary Figs. 20 and 21 show the time course of ALT, ALP and AST hepatic enzymes, creatinine and bilirubin for mice injected with $Ag_2S$ superdots (100 μL, 0.15 mg mL$^{-1}$). The serum concentrations of all biomarkers lie within healthy ranges. The values obtained 1 and 28 days after injection are shown in Fig. 5g–k both for mice intravenously injected with $Ag_2S$ superdots and for control mice. The hepatic enzymes show a discrete though not relevant increase at acute time points (24 h), while creatinine and bilirubin remain unaffected. Twenty-eight days after injection, the levels of all five metabolites are comparable between both groups. A more complete discussion of the time-course evolution of all the biomarkers is included in Supplementary Note 7. These results suggest a low in vivo toxicity for the $Ag_2S$ superdots, which is consistent with the previously reported good biocompatibility of $Ag_2S$ dots[41]. This is further supported by the low cytotoxicity of $Ag_2S$ superdots (see Supplementary Fig. 22) and in histological assays (Supplementary Fig. 23). This is a promising result, although it is important to remark that the clinical translation of $Ag_2S$ superdots requires further toxicity experiments including studies of the maximum tolerated dose, the clearance pathways and the effect of $Ag_2S$ superdots on physiology, metabolism, behavior, and cognition. Thus, such translation is not immediate.

## Discussion

Although $Ag_2S$ dots have shown excellent properties as NIR-II optical probes their potential use in preclinical applications is limited by their low brightness. Traditional synthesis routes lead to $Ag_2S$ dots with QY typically below 1%. Femtosecond laser pulses can enhance this QY by more than one order of magnitude, up to values above 10% (Fig. 2e), without affecting the spectral shape of the broadband emission centered at 1200 nm (Fig. 2d).

The dot-to-superdot transformation triggered by femtosecond laser pulses is strongly dependent on the experimental conditions. It requires the presence of silver NPs and chloroform molecules (Fig. 3b, c). Pulse widths longer than 100 fs lead to slow and inefficient dot-to-superdot transformation (Fig. 3d). We concluded that dot-to-superdot transformation is a sequential process (Fig. 3e): (i) femtosecond laser pulses induce the Coulomb explosion of silver NPs, (ii) silver atoms react with solvent molecules, and (iii) a protective shell of AgCl surrounds $Ag_2S$ dots. This protective shell (evidenced by electron microscopy, Fig. 1) drastically reduces surface-related non-radiative processes and leads to an increased brightness and a longer fluorescence lifetime (Fig. 2f).

In vivo NIR-II imaging experiments demonstrated the ability of $Ag_2S$ superdots for low-dose (administration and irradiation) preclinical imaging (Fig. 4b, c). We demonstrated how the optimization of the optical properties caused by femtosecond laser pulses improves both contrast and resolution of deep tissue in vivo images (Fig. 4g, h). An array of in vitro toxicity, biochemical and histological assays suggested a low toxicity for the $Ag_2S$ superdots (Fig. 5).

Our discovery, the ability of ultrafast laser pulses to improve the luminescence properties of NPs, also stimulates the development of new synthesis procedures that could benefit from the synergy between traditional chemical routes and light-matter interaction processes.

## Methods

**Chemical reagents**. Ethanol absolute, n-hexane (95%), Silver nitrate (99%), L-Cysteine (96%), sodium diethyldithiocarbamate (DDTC) (ACS reagent grade), $CHCl_3$ (99.6%), HS-PEG-COOH (2100 g mol$^{-1}$), and PBS tablets were purchased from Sigma-Aldrich (Germany) and used as received.

**Synthesis of Ag₂S dots**. $Ag_2S$ dots were prepared as follows: 3 mmol of silver nitrate was poured into a round bottom flask containing 10 mL of octadecylamine at 160 °C under gentle stirring and $N_2$ atmosphere. After 10 min, the mixture acquired a metallic blue color. At this stage, 1.5 mmol of L-cysteine was added to the mixture as sulfur source. As result, the color of the mixture turned from blue to black indicating the formation of the heterodimer (30 min). Then, the sample was cooled down and the product of the reaction was dispersed in 40 mL of $CHCl_3$. This dispersion was centrifuged at 21,000 g for 30 min and the precipitate was collected and dispersed in $CHCl_3$ at a concentration of 1 mg mL$^{-1}$.

**Ultrafast laser irradiation**. For ultrafast laser irradiation, we used a Ti:Sapphire amplified (Spitfire from Spectra-Physics) pumped by a Ti:Sapphire oscillator (Tsunami from Spectra-Physics) both operating at 808 nm. The amplifier provides pulses with a repetition rate of 1 kHz and tunable pulse widths in the 50–550 fs range by fine adjustement of the compensating gratings. The average power was controlled by using a set of polarizers and a λ/2 waveplate (AQWP10M-980 from Thorlabs). A 45 cm focal length lens was used to focus the irradiation beam into a hermetically closed quartz cuvette containing the dispersion of $Ag_2S$ dots in $CHCl_3$. The $Ag_2S$ concentration was set to 1 mg mL$^{-1}$ in all experiments. The focusing lens was mounted on a translation stage that allowed changing the lens-to-cuvette distance and, hence, the laser spot size in the solution. This, in turn, allowed us to change the laser power density while keeping the average laser power constant. Supplementary Fig. 24 shows, as a representative example, the laser spot size and the corresponding irradiation density as a function of the lens-to-cuvette distance for a fixed irradiation laser power of 0.6 W. The NIR-II emission was continuously registered during the irradiation process by a fiber-coupled spectrometer with enhanced sensitivity in the 900–1700 nm spectral range (Ocean Optics NIR-QUEST212). Real-time temperature measurements during irradiation were performed by means of a thermocouple placed into the solution. The distance between the thermocouple and the laser focal spot was set to 2 mm in order to avoid direct heating of the thermocouple by the 808 nm laser beam.

**Synthesis and characterization of Ag-NP-free Ag₂S dots (neat Ag₂S dots)**. The synthesis of the Ag-NP-free $Ag_2S$ dots was carried out by thermal decomposition of the precursor silver diethyldithiocarbamate (AgDDTC). The precursor was synthesized as follows: 0.025 mol of $AgNO_3$ were dissolved in 200 mL of bidistilled water produced from Milli-Q water. Later 0.025 mol of DDTC (diethyldithiocarbamate) were dissolved in 300 mL of bidistilled water and added to the above solution. The resulting yellow powder was filtered and dried at 60 in vacuum using a rotary evaporator. After that, 25 mg of AgDDTC was dispersed in 5 mL of 1-dodecanethiol and the mixture was stirred under vacuum for 30 min. After that, the solution was heated up to 200 °C for 1 h. After this time, the solution was cooled down naturally. When the temperature of the mixture reached 25 °C, 10 mL of ethanol was added and the solution was centrifuged at 10,000 rpm for 10 min. The supernatant was discarded and the precipitate collected in 10 mL of $CHCl_3$.

**Ligand exchange procedure**. To use the $Ag_2S$ superdots for in vivo imaging, we transferred them from $CHCl_3$ to water by a ligand exchange reaction between the octadecylamine and HS-PEG-COOH (Mw of 2100 g mol$^{-1}$). 2 mg of $Ag_2S$ superdots dispersed in 1 mL of $CHCl_3$ were mixed with 1 mg of HS-PEG-COOH and sonicated for 5 min to facilitate the ligand exchange. Afterwards, the $CHCl_3$ was gently evaporated until reaching a total volume of 500 μL. Then, 500 μL of absolute ethanol were added and the dispersion was sonicated for 5 min. This process was repeated four additional times in order to ensure the maximum removal of $CHCl_3$. The $Ag_2S$ superdots dispersed in 1 mL of absolute ethanol were then concentrated to a volume of 100 μL and transferred dropwise to 900 μL of PBS, prepared by dissolving a tablet in 100 mL of Milli-Q water.

**Scanning transmission electron microscopy (STEM)**. For the STEM studies, a JEOL ARM200 cF was used. This microscope is equipped with a Cs corrector in the condenser lens and an OXFORD INCA detector of 100 mm² for XEDS analysis. The experiments were carried out at 80 kV as the sample was beam sensitive to higher acceleration voltages. As a first step, sample stability was carefully checked in function of the different parameters of the microscope. To get the highest possible intensity in the XEDS detector without damaging the sample, we determined that the optimum conditions were achieved by using a condenser lens aperture of 50 micrometers and a spot size of 6. In these conditions, the probe current is well above 120 pA. The camera length for the STEM images was set to 6 cm so that the collection angles range from 90 to 370 mrad.

**Dynamic light scattering (DLS) and Z-potential determination**. The DLS and Z-potential measurements were performed using a Zetasizer Nano ZS instrument

(Malvern Instruments, U.K.). The accumulation time was determined automatically for each sample. The sample concentration was adjusted to obtain 3000 cps. The acquired data were processed using the software provided by Malvern (Zetasizer software v7.03).

**Quantum yield measurements**. The absolute photoluminescence QY was measured with a calibrated spectrofluorometer (Edinburgh Instruments, FLS920) equipped with an integrating sphere (Jobin-Yvon). A Xe lamp was used as the excitation source, filtered with a longpass filter (610 nm) and a monochromator (wavelength: 808 nm, bandwidth: 20 nm), and a liquid-nitrogen-cooled NIR photomultiplier tube (Hamamatsu, R5509-72) was used for detection. The QY was calculated by dividing the total number of emitted photons in the 900–1700 nm range by the total number of absorbed photons at 808 nm. A representative example of the excitation and emission spectra used for the determination of the QY is shown in Supplementary Fig. 7.

**Luminescence decay curves**. Luminescence decay curves were obtained by exciting the samples with an OPO oscillator pumped by a frequency doubled Nd: YAG laser (Lotis), which provides 8 ns pulses at a repetition rate of 10 Hz. The fluorescence intensity was detected with a Peltier-cooled photomultiplier tube with enhanced sensitivity in the NIR-II (Hamamatsu R5509-73). The contribution of scattered laser radiation was removed with two longpass filters (Thorlabs FEL850) and a high brightness monochromator (Andor Shamrock 320). The time evolution of the fluorescence signal was recorded and averaged by a digital oscilloscope (Le Croy Waverunner 6000).

**In vivo imaging**. NIR-II in vivo images were obtained in a custom-made NIR-II imaging system. A fiber-coupled diode laser operating at 808 nm was used as excitation source (LIMO30-F200-DL808). The illumination power density was controlled by adjusting the diode current. The anesthetized mouse was placed on a temperature-controlled plate set to 36 °C. The NIR-II fluorescence images were acquired with a Peltier-cooled InGaAs camera (Xenics Xeva 320) cooled down to −40 °C. Two longpass filters (Thorlabs FEL850) were used to remove the background signal generated by the scattered laser radiation. In addition, in vivo images were acquired using an additional 1100 nm longpass filter to ascertain the effect of further spectral filtering on the spatial resolution of the NIR-II images (see Supplementary Figs. 25 and 26).

In vivo experiments were approved by the regional authority for animal experimentation of Comunidad de Madrid and were conducted in agreement with the Universidad Autónoma de Madrid (UAM) Ethics Committee, in compliance with the European Union directives 63/2010UE and Spanish regulation RD 53/2013. Experiments were designed in order to use the minimal amount of animals, in accordance with the 3Rs ethical principle. No randomization or blind studies were performed. For this study, 15 CD1 female mice (8–14 weeks old, weighing 25–39 g) bred at the animal facility at UAM were used. Mice were anesthetized prior to the imaging experiments in an induction chamber with a continuous flow of 4% isoflurane (Forane, AbbVie Spain, S.L.U) in 100% oxygen until loss of righting reflex was confirmed and breathing rhythm was significantly slowed. Anesthesia was maintained throughout the experiments by means of facemask inhalation of 1.5% isoflurane and core body temperature was kept at 36 ± 1 °C, as measured with a rectal probe, using a heating pad.

To study the performance of $Ag_2S$ superdots as contrast agents in bioimaging when compared to other NIR-II probes, four mice were shaven and subcutaneously injected with 100 μL of a 1.5 mg mL$^{-1}$ NP ($Ag_2S$ superdots, $Ag_2S$ dots, $LaF_3$:$Nd^{3+}$ or SWNTs) dispersion in PBS.

To study the biodistribution of the $Ag_2S$ superdots, two additional mice received a small incision in the neck skin to expose the right jugular vein. A polyethylene tubing catheter was inserted 2 mm in the caudal direction to infuse 100 μL of solution containing 1.5 mg mL$^{-1}$ PEG-coated $Ag_2S$ superdots dispersed in PBS. Three hours after infusion, mice were euthanized by isoflurane overdose and organs (liver, spleen, heart, and lungs) were collected to obtain ex vivo NIR-II images.

One additional mouse was anesthetized in order to check the laser-induced thermal loading with and without fur. In this case, the temperature at the skin surface was measured with an infrared thermographic camera (Fluke iT10).

**In vivo biocompatibility studies**. Four additional mice were intraperitoneally injected with 300 μL of a 500 μg mL$^{-1}$ solution of $Ag_2S$ superdots in PBS. Four mice intraperitoneally injected with 300 μL of PBS were used as control group. The animals were housed in two separated cages in a room (23 ± 2 °C) maintained on a 12/12 h light/dark cycle with free access to food and water. Weight, food intake, and surface temperature, as measured by an infrared thermographic camera (FLIR E-40), were analysed in the awake mice on a daily basis for 2 weeks.

**In vivo subchronic toxicological studies**. Fifteen mice were intravenously injected with 100 μL of a 0.15 mg mL$^{-1}$ dispersion of $Ag_2S$ superdots and euthanized in groups of three at 24 h, 7, 14, 21, and 28 days post injection. Two additional groups ($n = 2$) were injected with PBS and euthanized after 24 h and 28 days. Serum ALT-

Alanine transaminase, ALP-Alkaline phosphatase, AST-Aspartate transaminase, creatinine, and total bilirubin were determined.

**In vitro cytotoxicity experiments**. Cytotoxicity of $Ag_2S$ nanodots was evaluated in HeLa (human cervical adenocarcinoma), NIH-3T3 (swiss mouse embryo fibroblasts) and IMR-90 (human lung fibroblasts) using the thiazolyl blue tetrazolium bromide (3-(4,5-dimethyl-thiazol-2yl)−2,5-diphenyltetrazolium bromide, MTT) assay. Briefly, cells in complete medium (DMEM with 10% FCS, 1 mM pyruvate, 2 mM glutamine, 100 U mL$^{-1}$ penicillin, 100 μg mL$^{-1}$ streptomycin, and 50 μg mL$^{-1}$ gentamicin) were seeded in 96-well culture plates at a density of $10^4$ cells per well. After 24 h of incubation (37 °C, 5% $CO_2$). Cells have been tested for mycoplasma contamination. $Ag_2S$ nanodots were added to cells at different concentrations up to 200 μg mL$^{-1}$ and further incubated for 48 h. Then, MTT solution was added for 4 h to the plate and afterwards, MTT reaction was stopped by adding a solution of dimethylformamide-SDS. Finally, the plate was gently shaken for 2 h to dissolve formazan crystals prior to measure 570/590 nm absorbance in an Appliskan (Thermo Scientific) plate reader. Corrected absorbance was transformed to percentage of cell viability using the following formula:

$$\%Cell\ viability = \frac{Abs\ sample}{Abs\ control} \times 100,$$

where Abs is the corrected absorbance at 570 nm after subtracting the absorbance at 590 nm.

Experiments were carried out four times and data were represented as a mean ± standard error.

**Reporting summary**. Further information on research design is available in the Nature Research Reporting Summary linked to this article.

## Data availability
Source data for all figures are available with the paper. All other data generated and analysed from this study are available from the corresponding authors upon reasonable request.

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

## Acknowledgements

Authors thank Dr A. Benayas (CICECO, U. Aveiro, Portugal), Prof G. Lifante and Prof J. García Sole (UAM) for helpful discussions. This work has been founded by Ministerio de Economía y Competitividad-MINECO (MAT2017-83111R and MAT2016-75362-C3-1-R) and the Comunidad de Madrid (B2017/BMD-3867 RENIM-CM) co-financed by European Structural and Investment Fund. D.M.-G. thanks UCM-Santander for a predoctoral contract (CT17/17-CT18/17). We thank the staff at the ICTS-National Centre for Electron Microscopy at the UCM for the help in the electron microscopy studies and C.M. at the beamline BL22-CLAESS of the Spanish synchrotron ALBA for his help in the XANES experiments. We also thank J.G.I at the Ultrafast Laser Laboratory at UCM for his help and fruitful discussion. Y.S. acknowledges the support from the China Scholarship Council (CSC File No. 201806870023). Additional funding was provided by the European Commission Horizon 2020 project NanoTBTech, the Fundación para la Investigación Biomédica del Hospital Universitario Ramón y Cajal project IMP18_38 (2018/0265). Ajoy K. Kar and Mark D. Mackenzie acknowledge support from the UK Engineering and Physical Sciences Research Council (Project CHAMP, EP/M015130/1). C. Jacinto thanks the financial support of the Brazilian agencies: CNPq (Conselho Nacional de Desenvolvimento Científico e Tecnológico) through the grants: Projeto Universal Nr. 431736/2018-9 and Scholarship in Research Productivity 1C under the Nr. 304967/20181; FINEP (Financiadora de Estudos e Projetos) through the grants INFRAPESQ-11 and INFRAPESQ-12; FAPEAL (Fundação de Amparo à Pesquisa do Estado de Alagoas) grant Nr. 1209/2016. H. D. A. Santos was supported by a graduate studentship from CNPq and by a sandwich doctoral program (PDSE-CAPES) developed at Universidade Autonoma de Madrid, Spain, Project Nr. 88881/2016-01.

## Author contributions

I.Z., M.L., D.M.-G., E.L.C., and J.R.R. fabricated the $Ag_2S$ dots. J.R.R., S.M., O.G.C., M.D.M., A.K.K., and D.J. designed and performed the ultrafast laser irradiation experiments. H.D.A.S., J.R.R., S.M., O.G.C., C.J. and D.J. characterized the optical properties of $Ag_2S$ dots and superdots. J.M.-H., C.M.S.J., and D.J. performed the quantum yield measurements. H.D.A.S., Y.S., J.L., E.X., N.F., L.M., B.delR. and D.J. designed and performed the in vivo experiments. I.C.-C., D.L.-A., J.L., and N.F. designed and performed the in vitro and in vivo biocompatibility experiments. J.R.R., B.del.R., and D.J. wrote the paper. H.D.A.S., I.Z.G., Y.S., J.L. and E.X. contributed equally to this work.

## Competing interests

The authors declare no competing interests.
