## [Peer Review File · Nature Communications]

Reviewers' comments:

Reviewer #1 (Remarks to the Author):

In this manuscript, the authors reported a novel strategy for preparing NIR-II Ag₂S superdots with high quantum yield (QY) and demonstrated their feasibility for in vivo depth-tissue imaging under a low excitation power density condition. This is a very interesting work that has demonstrated the marriage between chemical synthesis and optical processing techniques. The main finding of the study will provide the novel inspirations for bioprobe design. However, some issues should be further addressed. My comments are as follows:

1. In this case, the authors provided a good example for preparing bright Ag₂S superdots, whether this strategy is universality for other systems.
2. The authors described that the Ag₂S superdots exhibited an 80-fold increment in the QY of Ag₂S dots. As a control, the QY of the Ag₂S dots should also be assessed.
3. It is not clear whether 800 nm or 808 nm laser is used for in vivo imaging. For example, on page 14, lines 550-551, the author described that "NIR-II in vivo images.....A fiber-coupled diode 551 laser operating at 800 nm was used as an excitation source (LIMO30-F200-DL808)." However, on page 10, lines 375-377, the author described that "Fig. 4b shows the in vivo NIR-II fluorescence images obtained in each case for 808 nm illumination power densities ranging from 227 down to 0.3 mW/cm²."
4. In the caption of Figure 5, "time-resolved imaging" should be "time course of..."
5. The detailed information of statistics analyzes should be included for all figures (such as Figure 2e-f, Figure 5d-f), including sample size, the meaning of error bar, etc.
6. There are several minor errors, such as "Figure 3.- Required....(Page 9 line 333)", references 27, 30 and 32 (pages 18-19).

Reviewer #2 (Remarks to the Author):

This manuscript describes a kind of Ag₂S superdots that display bright NIR II fluorescence with an 80-fold enhanced quantum yield with respect to conventional Ag₂S dots. A protective shell is grown by femtosecond laser irradiation to reduce the non-radiative transitions and prevent quenching pathways. Such super-bright Ag₂S enabled minimum administration doses of <0.5 mg/kg when used in deep-tissue in vivo imaging. The topic of this femtosecond laser irradiation induced fluorescence enhancement is interesting. However, the explanation to the possible mechanism of this transformation is not convincing. More characterizations are needed to support authors' conclusion. The in vivo mouse vascular structure images obtained by intravenous injection of Ag₂S superdots were also not satisfying. The Ag₂S superdots were mainly circulating and gathering in the liver and spleen immediately after the injection of Ag₂S, demonstrating the limited application prospects of the Ag₂S superdots. Moreover, several major technical points need to be addressed as follow.

Questions related to the Ag₂S superdots include:

1. In Figure 1, the authors used STEM and EDS methods to verify the existence of Ag NPs in the as-synthesized Ag₂S dots and the transformation of Ag into AgCl. While the results were not strong enough to support that. I suggest some more characterizations including: XRD of Ag₂S dots and Ag₂S superdots to demonstrate the patterns of Ag, Ag₂S and AgCl in these samples; EXAFS (extended X-ray absorption fine structure) of Ag₂S dots and Ag₂S superdots to demonstrate the valence of Ag and existence of metallic Ag.
2. The authors stated that 'Laser pulses trigger the transformation of Ag NPs into AgCl particles', and the Ag core located eccentrically in Figure 1. Is there any explanation how can this discrete metallic Ag core generate an uniform AgCl shell around the whole NPs after ultrafast illumination.
3. By plotting the time derivative of intensity as a function of irradiation pulse energy, the authors suggested that 'the dot-to-superdot transformation is triggered by a two-photon absorption

process'. Is there any reference to support that?

4. It is better to update the literatures about this field, such as Nano Res., 2019, 12, 749; ACS Nano, 2017, 11, 1848 and Nano Res., 2019, 12, 273.

Questions related to the in vivo imaging include:

1. In Figure 4, the Ag₂S superdots and other NIR-II fluorescent probes were subcutaneously injected to compare their brightness, which is not appropriate because the injection depth is not controllable. The authors should do intravenous injection of these NIR-II fluorescent probes and compare their brightness in the vessels.
2. In Figure 5a, the Ag₂S superdots were mainly circulating and gathering in the liver and spleen, with very short blood half-life of 20 min. The vessel structure imaging quality in Figure 5 was also worse than previous work (Biomaterials, 2014, 35, 393-400). Does that mean the Ag₂S superdots are not suitable for in vivo imaging?
3. In the abstract, the authors stated that 'PEGylated Ag₂S superdots enable deep-tissue in vivo imaging at low excitation power densities (<1 mW/cm²) and minimum administration doses (<0.5 mg/kg)'. While in Figure 5a, the illumination power density is 50 mW/cm².
4. Histological studies and AST/ALT activity assays should be done to further verify the biocompatibility of Ag₂S superdots.

Reviewer #3 (Remarks to the Author):

In this manuscript, the authors have developed ultra-sensitive Ag₂S, which has a strong emission in the NIR-II window, by introducing an additional protective shell layer on the hydrophobic Ag core. The shelling process with femtosecond laser irradiation is simple and can efficiently reduce structural defects resulting in enhanced quantum yield of Ag₂S. Additionally, the physical and optical characterizations of Ag₂S dots before and after ultrafast laser irradiation have done satisfactorily. However, its application toward in vivo imaging seems to be very limited, which is not fully supported experimentally. In addition, the benefit of low dose/ low power in the NIR-II imaging is not fully demonstrated.

1. Optophysical analyses of Ag₂S dots are well presented in this manuscript. However, critical experimental details are not given. What is the temperature for all assays? What is the exact aqueous buffer or biological media used? How long was incubation? Was serum used? If so, what type and concentration? In the absence of serum, buffered salt solutions alone (i.e., cell culture media or PBS), immediately initiate signal transduction cascades turning on quiescence and/or apoptosis genetic programs.
2. Another caveat of this study is poor in vivo assay. Additional studies including PK/PD, such as single dose/repeated dose acute/chronic toxicity, are required to understand the possibility of using Ag₂S dots in the clinic. This assay may be beyond the scope of this manuscript, but I recommend performing at least biochemical analysis, biodistribution, and clearance study to prove the potential use of Ag₂S in animal models.
3. What is the maximum penetration depth of Ag₂S super dots in the NIR-II imaging setup? This can be simply performed using a tissue phantom.
4. Two longpass filters (Thorlabs FEL850) were used in this study. For better resolution, NIR-II imaging should be carried out with a longer wavelength filter such as 1100 nm LP.
5. This study is missing an important control. ICG will be a good model, although it uses a tail of emission for NIR-II imaging. Comparison should be done in serum containing media.
6. Page 3, "The combination of already available Ag₂S dots and current image acquisition technologies allows for imaging depths of few centimetres whereas clinical applications require penetration depths one order of magnitude larger." This sentence needs to be supported by reference(s). As far as I know, the penetration depth of NIR-II imaging is limited to a few millimeters, not centimeters in the body. Although this is argued in Introduction, no supporting data is presented to support this statement as well.
7. Page 3, "we have developed a novel methodology that permits a 80-fold increment in the quantum yield of Ag₂S dots." Although the chemical reaction for NPs is the same, the methodology of laser irradiation is interesting. I was wondering if this photo radiation induced shelling method could be applied to the other nanocrystals to enhance their QY.

Reviewer #1

R1.1. In this case, the authors provided a good example for preparing bright Ag₂S superdots, whether this strategy is universality for other systems.

The photochemical reaction at the basis of the formation of the AgCl protective shell in superdots seems to depend only on the presence of Ag nanoparticles in the presence of chloroform. Thus, we think that the same process can be applied to other types of nanoparticles if they are dispersed in chloroform and in the presence of Ag nanoparticles. This possibility has not been explored at this point, but we are firmly convinced that it will be worth exploring for the fabrication, as an example, of superbright lanthanide-doped nanoparticles. In this kind of fluorescent nanoparticles, the improvement in their surface quality has been already demonstrated to provide remarkable enhancements in their brightness (Lei, J., *et al.*, *Materials Today Nano* 8 (2019): 100055) but the possibility of growing such a protective shell by using ultrafast photochemistry remains unexplored. This fact is now stated in this revised version.

R1.2. The authors described that the Ag₂S superdots exhibited an 80-fold increment in the QY of Ag₂S dots. As a control, the QY of the Ag₂S dots should also be assessed.

We would like to note that in our original submission we included the QY of Ag₂S dots. Note that in **Figure 2e**, the QY for a time irradiation of $t = 0$ min corresponds to the QY of as-synthesized Ag₂S dots in chloroform. In our original submission, we also showed the experimental procedure followed to determine the QY of Ag₂S dots in PBS but we did not include the experimental data from which the QY of these samples (Ag₂S dots and superdots) were calculated. These data have been now included in **Section S5** of the Supplementary Information, which now incorporates the pump (with and without sample) and luminescence spectra used for the determination of the QY of Ag₂S dots and superdots in CHCl₃ and in PBS. The new **Supplementary Fig. 7** is included in this response letter as **Figure RL1**.

Figure RL1. Illustrative examples of the spectra used for the determination of the QY. Data correspond to dispersions of Ag₂S dots and superdots in CHCl₃ in PBS. The spectra of the excitation light (808 nm) is measured with and without sample. From the difference between these spectra, the number of absorbed photons is calculated in each case. The number of emitted photons generated by each sample is calculated from the emission spectra recorded in presence of the sample. Then, the QY is calculated by dividing the number of emitted photons by the number of absorbed photons. All the emission spectra are corrected by the system response.

R1.3. It is not clear whether 800 nm or 808 nm laser is used for *in vivo* imaging. For example, on page 14, lines 550-551, the author described that “NIR-II *in vivo* images.....A fiber-coupled diode 551 laser operating at 800 nm was used as an excitation source (LIMO30-F200-DL808).” However, on page 10, lines 375-377, the author described that “Fig. 4b shows the *in vivo* NIR-II fluorescence images obtained in each case for 808 nm illumination power densities ranging from 227 down to 0.3 mW/cm².”

We apologize for this mistake. The excitation wavelength used for *in vivo* imaging was 808 nm. We have modified the text accordingly.

R1.4. In the caption of Figure 5, “time-resolved imaging” should be “time course of...”

The caption of Figure 5 has been modified accordingly.

R1.5. The detailed information of statistics analyzes should be included for all figures (such as Figure 2e-f, Figure 5d-f), including sample size, the meaning of error bar, etc.

The error bars associated to QY measurements that are included in **Figure 2e** were determined from the statistical analysis of repeated measurements. Note that as a result of the re-analysis of experimental data we found experimental uncertainties lower than those included in our original version. Indeed, the relative uncertainties of our QY measurements are of the order of 1% so they are not observable in **Figure 2c**. The error bars in **Figure 2f** correspond to the standard deviations as obtained by measuring and analysing 10 decay curves for each sample. In Figure 4, the experimental data have no error bars as these data were obtained from the analysis of the *in vivo* images included in **Figure 4b**, which are the fluorescence images obtained after subcutaneous injection of different NIR-II probes. These experiments were performed on a single mouse per NIR-II probe in accordance to our Ethics approval, preventing us from performing any statistical analysis. Please note that, as discussed in point R2.5 of this response letter, the purpose of these experiments was to demonstrate that Ag₂S superdots allow *in vivo* imaging at signal-to-noise ratios of orders of magnitude above those achievable with other NIR-II nanoprobe, so that statistical analysis did not justify the use of additional animals. This fact is clearly stated in the revised version.

R1.6. There are several minor errors, such as “Figure 3.- Required....(Page 9 line 333)”, references 27, 30 and 32 (pages 18-19).

We thank the reviewer for their careful reading of our manuscript. All these typos have been amended.

Reviewer #2.

R2.1. In Figure 1, the authors used STEM and EDS methods to verify the existence of Ag NPs in the as-synthesized Ag₂S dots and the transformation of Ag into AgCl. While the results were not strong enough to support that. I suggest some more characterizations including: XRD of Ag₂S dots and Ag₂S superdots to demonstrate the patterns of Ag, Ag₂S and AgCl in these samples; EXAFS (extended X-ray absorption fine structure) of Ag₂S dots and Ag₂S superdots to demonstrate the valence of Ag and existence of metallic Ag.

To provide a complete chemical and structural characterization of the samples, we have included additional information like X-ray diffraction (XRD) patterns, X-ray absorption near edge structure (XANES) spectra, energy-dispersive X-ray (EDS) spectra and high-angle annular dark-field scanning transmission electron microscopy (HAADF-STEM) images of the Ag₂S dots and superdots.

Figure RL2. (a) XRD pattern of the as synthesized Ag₂S dots sample. The red lines represent typical reflections of monoclinic Ag₂S phase, whilst green lines represent the reflection positions of cubic Ag. (b) XANES spectra of Ag₂S dots (red) and Ag₂S used as reference (blue) and Ag foil used as reference (black). (c) EDS spectrum of Ag₂S dots. (d) EDS elemental mapping of Ag₂S, silver (yellow), sulphur (red), the inset in the bottom left is the resulting image after merging S+Ag and the inset in the bottom right is the HAADF-STEM image from which the EDS map was collected. The scale bar is 10 nm.

Figure RL2 (a) of this response letter shows the XRD pattern of the Ag₂S dots sample, where diffraction peaks at 34.385° (-121) and 36.556° (112), which can be assigned to the monoclinic phase of Ag₂S (JCPDS card No. 14-0072; lattice constants: a = 4.229 Å, b = 6.931 Å, c = 7.862 Å), can be observed. Furthermore, in this figure, we see reflection peaks at 38.116°, 44.277°, 64,72 and 77,40 that can be assigned to cubic phase of

Ag (111), (200), (220) and (311) (JCPDS card No. 04-0783; lattice constants: $a = 4.0862 \text{ \AA}$). Therefore, the information extracted from the XRD spectrum would indicate the presence of two crystalline phases in the sample, corresponding to monoclinic Ag_2S and cubic Ag phases, respectively.

Following the suggestion of the reviewer, we have studied the oxidation state of the Ag atoms of this sample. For that purpose, XANES (also referred to as NEXAFS) experiments were performed. To analyse the oxidation state of Ag atoms, we reconstructed the spectrum by a linear combination fitting procedure. XANES experiments performed at the Ag K-edge indicate that the 25% out of the total Ag content is metallic Ag, while the counterpart (75%) is Ag^+ forming part of Ag_2S as shown in **Figure RL2 (b)**. This result is in good agreement with the EDX spectrum of the Ag_2S dots shown in **Figure RL2 (c)**. This spectrum indicates the atomic percentage of sulphur in the sample is around 26% while the atomic percentage of silver is 74% (within this percentage there is no difference between Ag^0 or Ag^+). If we consider the Ag_2S stoichiometry, each sulphur atom should be bound to 2 ions of Ag^+ , so from the total content of Ag ions (74%), 52% would be Ag^+ ions forming Ag_2S and the remaining 22 % would correspond to metallic Ag.

Figure RL2 (d) shows an EDS-mapping analysis of Ag_2S dots. In this micrograph, one can observe the presence of regions enriched in silver, while the presence of sulphur is homogeneously distributed around the structure of the nanoparticles except in those regions where there is a clear enrichment of Ag. The regions enriched in Ag atoms match with those electrodense regions observed in the HAADF-STEM inset. HAADF-STEM can be used to distinguish variations in the atomic number of atoms in the sample (Z-contrast), making it useful to identify small areas of an element with a high Z in a matrix of material with a lower Z. In this case, the metallic silver region would exhibit higher Z-contrast than the Ag_2S regions. In summary, all these results indicate clearly the structural and chemical composition of the $\text{Ag}_2\text{S}/\text{Ag}$ dots as the product of two different phases (metallic Ag and Ag_2S) integrated in each nanoparticle. This is clearly stated in the revised version of our manuscript within the Supporting Information file.

Figure RL3. (a) XRD pattern of Ag_2S superdots. (b) XANES spectra of Ag_2S superdots (orange) and Ag_2S used as reference (blue), AgCl used as reference (red) and Ag foil used as reference (black). (c) EDS spectrum of Ag_2S superdots. (d) EDS elemental mapping of Ag_2S superdots, silver (yellow), sulphur (red) and chlorine (blue). The inset in the bottom right is the resulting image after merging the S, Ag and Cl HAADF-STEM images. The scale bar is 10 nm.

Figure RL3 (a) shows the XRD pattern of the Ag_2S superdots. When the superdot sample was analysed by XRD, we observe the presence of reflections that can be attributed to monoclinic Ag_2S (JCPDS card No. 14-0072; lattice constants: $a = 4.229 \text{ \AA}$, $b = 6.931 \text{ \AA}$, $c = 7.862 \text{ \AA}$). Particularly, the peaks at 34.385° (-121) and 36.556° (112) can be assigned to the monoclinic Ag_2S . In addition, XRD pattern shows reflections at 38.116° , 44.277° , and 64.72° that correspond to cubic Ag phase (111), (200) and (220) (JCPDS card No. 04-0783; lattice constants: $a = 4.0862 \text{ \AA}$) that could be attributed to the presence of cubic metallic Ag . Finally, some tiny reflections at 32.312° (200), 46.138° (220) and 55.262° (311) could be attributed to cubic AgCl (JCPDS file 31-1238 ; lattice constant $a = 5.549 \text{ \AA}$). This result confirms the presence of an additional phase of AgCl in the superdot sample after the ultra-fast laser illumination.

In addition, the XANES (**Figure RL3 (b)**) spectrum of a sample of superdots was reconstructed by linear combination fitting procedure using three components; Ag_2S , metallic Ag and AgCl . The result of this analysis indicated that after the ultra-fast laser illumination, the percentage of Ag^+ that stemmed from Ag_2S phase remained around 70%. However, the percentage of Ag^0 was reduced from 22% to 14%, such a reduction was accompanied with the apparition of a new component of Ag^+ as AgCl in a percentage close to 16%, indicating the transformation of Ag^0 into Ag^+ as a result of the laser illumination. The EDS analysis of Ag_2S superdots (**Figure RL3 (c)**) indicates the presence of chlorine atoms in the sample. In fact the overall atomic composition of the sample was; silver 67%, sulphur 21% and chlorine 12%. Considering the Ag_2S

stoichiometry, the Ag forming Ag_2S phase would be 42% out of 67%. 12% of Ag atoms would form AgCl, since the atomic percentage of Cl is 12% and the remaining 13% of Ag atoms would form Ag^0 . These results are very close to the observed in the XANES experiments.

Finally, EDS-mapping analysis of the nanoparticles was carried out to see the localization of each atom. The results (**Figure RL3 (d)**) show that the distribution of the Ag atoms is not homogenous, with some regions enriched in Ag atoms. In addition, the S atoms seems to be homogeneously distributed in the NPs with the exception of those regions enriched in Ag atoms, where the presence of S atoms is conspicuously reduced. Finally, the Cl atoms are homogeneously distributed throughout the whole nanostructure and it would indicate the presence of a AgCl phase in the whole nanoparticle.

In summary, these results demonstrate that the as-synthesized dots are composed of two phases, a monoclinic Ag_2S phase and a cubic Ag phase. Further, when these NPs are irradiated with an ultrafast laser, the chemical composition and structure of the nanoparticles changes and the resulting nanoparticles exhibit a new AgCl phase that would cover the NPs forming a thin shell. This thin shell can be observed in Figure 1i of the main text. In fact, when we compare the high resolution TEM micrograph of a Ag_2S dot and a Ag_2S superdot (Figure 1b and 1i) we see that Ag_2S superdots exhibit an inorganic layer around the nanoparticle that does not appear in the Ag_2S dots used as precursors.

R2.2. The authors stated that ‘Laser pulses trigger the transformation of Ag NPs into AgCl particles’, and the Ag core located eccentrically in Figure 1. Is there any explanation how can this discrete metallic Ag core generate an uniform AgCl shell around the whole NPs after ultrafast illumination.

The silver nanoparticles that triggers the formation of the AgCl protective shell are the Ag nanoparticles that coexist, as side-products, with the as synthesized Ag_2S nanoparticles in the colloidal solution. The **Figure RL4 (a)** shows the product of the synthesis without applying any cleaning procedure. In this image, we can observe the presence of electrodense NPs that coexist with other, less electrodense NPs. A magnification image by high resolution TEM (HR-TEM) (**Figure RL4 (b)**) demonstrates the polycrystalline nature of one of these electrodense NPs, where we can see lattice fringes of 0.2 nm and 0.23 nm that would correspond to the 200 and 111 planes of cubic Ag. Furthermore, HAADF-STEM image (**Figure RL4 (c)**) of these samples highlight the difference in Z-contrast between Ag NPs (label with red arrows) and $\text{Ag}_2\text{S}/\text{Ag}$ NPs. Finally, the EDS elemental analysis of one NP that showed high Z-contrast demonstrates that they are composed of Ag.

Figure RL4. (a) TEM image of the as synthesized Ag₂S/Ag nanoparticles. (b) HR-TEM of a single electrodeposited nanoparticle, showing its polycrystalline structure with lattice fringe 0.20 and 0.23 nm. (c) HAADF-STEM image of the as synthesized Ag₂S/Ag nanoparticles. (d) EDS elemental analysis obtained from a high Z-contrast nanoparticle labeled with the red crosshair.

In summary, this information would demonstrate the presence of Ag NPs as a side product of the reaction. This has been clearly stated in the revised version of the manuscript and a new figure (Supplementary Fig. 1) has been included in the revised supplementary material.

Upon illumination with ultrafast laser at intensities below the damage threshold (9 W/cm²), the Ag core remains unaffected as can be observed in the images included in **Figure 1** in the main text. An arising question is why the multiphoton excitation and subsequent coulomb explosion takes place only in the colloidal Ag NPs and not in the Ag cores. A possible explanation is that the Ag core does not show any plasmonic resonance in the visible domain due to the presence of Ag₂S surrounding it. The spectral location of the plasmon resonance of Ag nanoparticles strongly depends on the dielectric constant/refractive index of the surrounding medium. The very different dielectric constants/refractive indices of chloroform and Ag₂S could be behind this effect. By contrast, the Ag nanoparticles exhibit a plasmon resonance at 425 nm, which favours nonlinear multiphoton absorption. In addition, that would result in an enhancement of the local electric field during the laser illumination that in turn facilitates the electron stripping from the Ag nanoparticles, enabling the ionization of the Ag nanoparticles and their subsequent Coulomb explosions, in a similar way as previously described by the Castleman group [Chem. Phys. Lett. 229, 333 (1994)]

Note that the crucial role of colloidal Ag nanoparticles was demonstrated by the experimental evidence included in **Figure 3c**: when Ag nanoparticles are eliminated from the solution there is no dot-to-superdot transformation (even in the presence of the Ag core). This fact is explained with more detail in the revised version.

R2.3. By plotting the time derivative of intensity as a function of irradiation pulse energy, the authors suggested that ‘the dot-to-superdot transformation is triggered by a two-photon absorption process’. Is there any reference to support that?

At this point, we apologize for our lack of clarity when discussing these data. There is not any reference supporting this as it is, instead, a conclusion we extracted from our experiments. To investigate the multiphoton origin of the dot-to-superdot transformation, we have studied the dependence of the dot-to-superdot transformation speed on the laser pulse energy. To do so, we measured the time evolution of NIR-II fluorescence during irradiation with different laser pulse energies. For each pulse energy, the time evolution of the NIR-II luminescence was measured in its linear regime that, according to what is explained in the main text, corresponds to an initial stage of dot-to-superdot transformation in absence of laser-induced damage (see **Figure RL5 (a)** of this response letter). The slope of each curve dl/dt was determined and plotted versus pulse energy in a log-log plot (**Figure RL5 (b)** of this response letter). The experimental data included in this last graph can be nicely fitted to a linear function in the log-log plot with a slope close to 2. This means that the slope dl/dt follows a power law behavior with an exponent equal to 2, i.e. $dl/dt \propto (E_p)^2$.

We can theoretically relate the time evolution of the fluorescence intensity and the rate of dot-to-superdot transformation by writing the fluorescence intensity as:

$$I = k_{sd} \cdot N_{sd} + k_d \cdot N_d \quad \text{(RLEQ1)}$$

where N_{sd} (N_d) is the population of superdots (dots) in the solution, and k_{sd} (k_d) is a constant that depends on different parameters including the fluorescence quantum yield and the molar extinction coefficient of the

superdots (dots), and on the optical collecting efficiency of the optical system used for luminescence detection. Then, time derivation of expression RLEQ1 leads to:

$$dI/dt = k_{sd} \cdot dN_{sd}/dt + k_d \cdot dN_d/dt \quad \text{(RLEQ2)}$$

As $dN_d/dt = -dN_{sd}/dt$, i.e., the rate of dot annihilation is the same as the rate of super-dot creation, then:

$$dI/dt = (k_{sd} - k_d) dN_{sd}/dt \quad \text{(RLEQ3)}$$

Thus, for initial instants of the ultrafast laser-induced dot-to-superdot transformation, the rate at which the fluorescence intensity increases with time (dI/dt) can be considered proportional to the number of dot-to-superdot transformations per unit time, i.e. to dN_{sd}/dt . At the same time, the dot-to-superdot transformation per unit time depends on the pulse energy (E_p). As we found from the experimental data dI/dt follows a power law behavior with an exponent equal to 2, i.e. $dI/dt \propto (E_p)^2$

$$dI/dt \propto dN_{sd}/dt \propto (E_p)^2 \quad \text{(RLEQ4)}$$

Therefore, we conclude that the dot-to-superdot transformation is triggered by a two-photon process. This fact is explained in more detail in the revised version. The experimental curves of NIR-II intensity versus time obtained for different pulse energies have also been included in the revised version.

The two-photon behaviour experimentally observed is in agreement with the proposed mechanism for the dot-to-superdot transformation. As explained in the main text, the dot-to-superdot transformation is triggered by the coulomb explosion of the Ag NPs (side products of the chemical synthesis) that coexist with Ag_2S dots in the colloidal solution. As can be observed in **Figure 2b** in the main text and in Figure S3 in the Supplementary Information, the Ag nanoparticles show a plasmonic resonance at around 450 nm. Therefore, an efficient excitation of the plasmonic resonance, (leading to coulomb explosion) requires the absorption of two pump photons. Thus, a quadratic dependence of the dot-to-superdot transformation on the pulse energy could be expected. All this discussion, together with the new data included in Supplementary Figure 9, has been included in our revised version.

Figure RL5. (a) Time dependence of the NIR-II fluorescence generated by a colloidal solution of Ag_2S dots in chloroform when being irradiated with 50 fs, 800 nm laser pulses of different energies. **(b)** The time derivative of intensity as a function of irradiation pulse energy (E_p). The slope of the curve indicates that η is directly proportional to $(E_p)^2$ so that it is a two-photon process.

R2.4. It is better to update the literatures about this field, such as Nano Res., 2019, 12, 749; ACS Nano, 2017, 11, 1848 and Nano Res., 2019, 12, 273.

The following references have been included in our revised version:

- “Infrared fluorescence imaging of infarcted hearts with Ag₂S nanodots” Dirk H. Ortgies et al. Nano Research 12, 749 (2019).
- “Size-Dependent Ag₂S Nanodots for Second Near-Infrared Fluorescence/Photoacoustics Imaging and Simultaneous Photothermal Therapy” T. Yang et al. ACS Nano 11, 1848 (2017).
- “A theranostic agent for cancer therapy and imaging in the second near-infrared window” Z. Ma et al. Nano Research. 12, 273 (2019).

R2.4. In Figure 4, the Ag₂S superdots and other NIR-II fluorescent probes were subcutaneously injected to compare their brightness, which is not appropriate because the injection depth is not controllable.

As pointed out by the referee, the depth of a subcutaneous injection is not fully controllable even when following the exact same protocol. In our experiments, special care was taken to ensure that the dispersions containing the NIR-II fluorescent nanoparticles were injected between the inner face of the skin (dermis + epidermis + hypodermis) and the muscular wall.

In these conditions, the possible variation in the injection depth is caused by the non-homogeneous thickness of the skin. According to the literature, the average thickness of the skin of a mouse is close to 400 μm (Azzi, Lamia, *et al.*, Journal of Investigative Dermatology 124.1 (2005): 22-27.). The variation in skin thickness of mice is close to 80 μm. Based on this thickness variation we can estimate the magnitude of the fluctuation in the detected fluorescence intensity. The subcutaneously injected NIR-II probes generate a fluorescence intensity I_0 . According to the Lambert-Beer law, the fluorescence intensity that comes out from the skin and, therefore, detected by the fluorescence camera, I_{det} , is given by:

$$I_{det} = I_0 \cdot \exp(-d \cdot \alpha_{ext,skin}) \quad \text{(RLEQ6)}$$

where $\alpha_{ext,skin}$ is the optical extinction coefficient of skin that, in the NIR II, is close to 25 cm⁻¹. In equation (RLEQ6) $d = 400 \mu\text{m}$ is the skin thickness. Then, a change in the skin thickness of $\Delta d = 80 \mu\text{m}$ will cause a change in the detected intensity, ΔI_{det} , that can be estimated by:

$$\Delta I_{det} = -I_0 \cdot \exp(-d/\alpha_{ext,skin}) \cdot (d/\alpha_{ext,skin}) = -I_{det} \cdot (\Delta d \cdot \alpha_{ext,skin}) = -I_{det} \cdot 0.2 \quad \text{(RLEQ7)}$$

This indicates that uncertainty in the injection depth due to skin thickness inhomogeneities is expected to be close to 20%. This variation is much smaller than the differences observed between the fluorescence intensities observed experimentally.

When comparing the *in vivo* images obtained after subcutaneous injection of the different NIR-II probes, we observe that the contrast of the fluorescence images obtained with superdots is several orders of magnitude larger than those obtained with the other NIR-II fluorescent probes. Therefore, the possible variations in the injection depth cannot explain the much better performance of superdots and that this is caused by their superior brightness. All these calculations are now included in the revised version. It is also stated clearly that the large improvement in the image contrast obtained with superdots cannot be attributed to changes in the injection depth but arise from the superior brightness of superdots.

By using (RLEQ7), it is also possible to estimate the variation in the injection depth that would be necessary to provide a change in the detected intensity of 2 orders of magnitude, which, as an example, is the ratio between the image contrasts obtained by using superdots and Nd:NPs (see Figure 4 in the main text). We found that such difference in the image contrast would occur if the Nd:NPs were injected 4 cm deeper than

the superdots. Such a difference in injection depths is not compatible with our experimental conditions. All this considered, we believe Figure 4 shows valuable data and that the conclusions extracted from it are valid, even when considering the error that could arise due to slightly different injection depths.

R2.5.- The authors should do intravenous injection of these NIR-II fluorescent probes and compare their brightness in the vessels. In Figure 5a, the Ag₂S superdots were mainly circulating and gathering in the liver and spleen, with very short blood half-life of 20 min. The vessel structure imaging quality in Figure 5 was also worse than previous work (Biomaterials, 2014, 35, 393-400). Does that mean the Ag₂S superdots are not suitable for in vivo imaging?

The reviewer states that the short blood half-life (20 minutes) of the Ag₂S superdots limits their potential application for *in vivo* imaging. At this point, we would like to note that the major point of this work is the demonstration of a new route to significantly enhance the brightness of Ag₂S dots, transforming them into superdots. This process does not change significantly their size while keeping their chemical composition. The circulation time of NPs with the same concentration and size it is known to be determined by the surface coating. Ag₂S superdots could be coated with molecules (such as branched PEG) that confer them long *in vivo* circulation times. Albeit this can sometimes be necessary (i.e blood flow imaging, passive tumor targeting), depending on the model chosen and the scope of the study it may be other situations in we would rather be interested in minimizing the time in which the nanoprobe reaches the specific target, usually a cell type of a particular tissue. This would be the case when looking to minimize potential off-target associated toxicity to organs (see for instance C. Li *et al.*, Biomaterials 35, 393 (2014)). Nevertheless, applying surface coatings to the NPs to optimize circulation times is out of the scope of this work, which, as stated before, focuses on demonstrating the ability of femtosecond laser pulses to trigger the dot-to-superdot transformation. This fact is clearly stated in our revised version.

The reviewer also asks about the potential use of our Ag₂S superdots for *in vivo* visualization of vessels, particularly about a comparative evaluation of the resolution achievable with Ag₂S dots and superdots. To answer this point, we have performed additional experiments using Ag₂S superdots for high-resolution vessel visualization. Two CD1 female 3 month-old mice were subjected to an intravenous tail injection with 100 μ L of 0.15 mg/mL dispersions of Ag₂S dots and superdots in PBS. 30 seconds after injection, magnified fluorescence images of the right limbs of both mice were obtained, as shown in **Figure RL6** of this response letter. Both images were obtained under the same experimental conditions: 50 mW/cm² excitation power density at a wavelength of 808 nm, emission signal filtered by two 850 nm and one 1100 long pass filters and a camera exposure time of 5 seconds. As it can be observed, Ag₂S superdots enable blood vessel imaging at a much higher contrast than Ag₂S dots thanks to their superior emission brightness. Such improvement allows not only better imaging of femoral artery but also to the visualization of additional vessels. Figure RL5 also includes the intensity profiles, obtained with both Ag₂S dots and superdots, along the dashed lines included in the fluorescence images and that correspond to a cross section of a secondary vessel. The presence of the secondary vessels is clearly evidenced by Ag₂S superdots while they are impossible to observe when using Ag₂S dots.

Figure RL5 evidences that both dots and superdots provide a clear image of the major blood vessels in the mice leg shortly after injection but only Ag₂S superdots allow visualizing narrower blood vessels. Indeed, the analysis of the intensity profile of the secondary vessel obtained with Ag₂S superdots reveal that they provide a sub-200 nm spatial resolution. All this discussion as well as the new data obtained have been included in the revised version.

Figure RL6.- NIR-II fluorescence images of the left leg of CD1 mice obtained 15 seconds after injection of 100 μL of Ag_2S superdots (a) and Ag_2S dots (b) at a concentration of 0.15 mg/mL. (c) Fitted Gaussian function of the cross-section intensity profiles of Ag_2S superdots and Ag_2S dots obtained along the whole paw. (d) Gaussian function fitting of cross-section intensity profile of Ag_2S superdots obtained from the dashed line depicted in (a).

R2.6. In the abstract, the authors stated that ‘PEGylated Ag_2S superdots enable deep-tissue *in vivo* imaging at low excitation power densities (<1 mW/cm²) and minimum administration doses (<0.5 mg/kg)’. While in Figure 5a, the illumination power density is 50 mW/cm².

The reviewer is right to point out that in our original manuscript we stated that Ag_2S superdots allowed us to obtain *in vivo* images at very low excitation power densities but provided no evidence beyond the images obtained for subcutaneously injected NPs. We do agree with the reviewer that the *in vivo* images in that case are not very representative due to the proximity of the injection site to the skin surface. To provide an accurate and realistic number of the minimum power density that we can use for *in vivo* imaging with Ag_2S superdots, we designed a new experiment based on the images obtained after intravenous injection of the particles.

The intravenous injections of Ag_2S dots was performed as previously described in question R2.5 of this response letter. Fluorescence *in vivo* images using an 808 nm laser at different power densities (between 10 and 200 mW/cm²) were obtained using a camera exposure time of 5 seconds and the signal-noise ratio (SNR, expressed in dB) was calculated in each case. The *in vivo* NIR-II fluorescence images included in **Figure RL7** reveal that the contrast of the image decreases with the illumination power density, as could be expected. It is also evidenced that clear images can be obtained even at illumination power densities as low as 10 mW/cm². Data included in Figure RL6 also indicates that the SNR of the fluorescence image decreases linearly with the illumination power density used. Even if 50 mW/cm² allows *in vivo* imaging with high SNR, our data suggests that we could theoretically achieve a SNR value of nearly 9 dB if we extrapolate the SNR

for an excitation power density of 1 mW/cm² of intensity. Considering these new data, we have modified the abstract to read “PEGylated Ag₂S superdots enable deep-tissue *in vivo* imaging at low excitation power densities (<10 mW/cm²) and minimum administration doses (<0.5 mg/kg)”. The data in **Figure RL7** are included in Figure 4 in the revised version of our manuscript.

Figure RL7. (a) Fluorescence images of Ag₂S superdots **accumulated in the liver** obtained at three different excitation power densities with an 808 nm laser (200, 140, and 10 mW/cm² from left to right). **(b)** Power density dependence of SNR calculated from the *in vivo* NIR-II fluorescence at different excitation power densities (10-200 mW/cm²).

R2.7. Histological studies and AST/ALT activity assays should be done to further verify the biocompatibility of Ag₂S superdots.

To test the long-term biocompatibility of Ag₂S superdots, we performed a 28-days subchronic toxicological experiment complemented with histological analyses regarding splenic, hepatic, and renal tissue samples. CD1 mouse strain is optimal to conduct toxicological studies due to their increased fidelity to represent the genetic heterogeneity background of a population if compared to another typical outbred strains usually used for imaging studied such as C57/BL 6 mice. A total of 19 CD1 1-2 month-old female mice under 2% of isofluorane-induced anesthesia were intravenously injected through the retro-orbital sinus with a solution of Ag₂S superdots in sterile-PBS, with an injection volume corresponding to that used for *in vivo* imaging (100 μl of a dispersion at a concentration of 0.15 mg/mL). As a control, mice were injected with the same volume of PBS. 5 groups (n=3) were injected with Ag₂S and euthanized after 24 h, 7 days, 14 days, 21 days and 28 days post-injection. To include an indicative control of both potential early acute and late sub-chronic stages, 2 additional groups (n=2) were injected with PBS and euthanized after 24 h and 28 days respectively. We choose to reduce the number of animals in these groups, as well as to limit the study to the acute and last stage after injection in order to minimize the number of animals necessary for the study. Mice previously anesthetized by isofluorane anesthesia (5%) were euthanized by beheading, the blood (1-1.2 mL) was collected and serum was obtained after coagulation and centrifugation (1 hours at room temperature and 2 hours at 4 °C followed by 15 minutes of 10,000g centrifugation at 4 °C). All samples were stored at -80 °C and biochemical determinations of colorimetric end point and 2 point kinetic curves assays were performed by in collaboration with the laboratory of clinical analysis (Laboratorio Alemany, Madrid, Spain). Toxicological assays were performed to ascertain possible changes in hepatic (ALT-Alanine transaminase, ALP-Alkaline

phosphatase, and AST-Aspartate transaminase), renal (creatinine) and hemolytic profiles (total bilirubin), and a toxicokinetic profile was obtained for Ag₂S treated mice and sacrificed at increasing time points up to 28 days.

Figure RL8 (a)-(e) summarize the ALT, ALP, AST, creatinine and bilirubin content for mice intravenously injected with Ag₂S superdots and control mice as obtained 1 and 28 days after injection. As can be observed, the hepatic enzymes show a discrete though not significant increase at acute time points (24 h), meanwhile the levels of creatinine and total bilirubin seem to remain unaffected. Importantly, the level of all 5 metabolites studied at 28 days after Ag₂S superdots injection are comparable to that of the control groups, supporting the notion that they are suitable for *in vivo* studies with a modest effect in the hepatic enzymes during the acute phase of the treatment. Data of **Figure RL8** have been included in the revised version within the Figure 5g-k. The time course profiles have been added to the Supplementary Material (Supplementary Figures 19 and 20) together with a detailed description of the experimental procedure. The comparison of results obtained for injected and control mice have been added to the main text (Figure 5).

Figure RL8. (a)-(c) Serum concentration of hepatic enzymes for mice intravenously injected with Ag₂S superdots and for control mice as obtained 1 and 28 days after injection. (d)-(e) Serum concentration of creatinine and bilirubin corresponding to mice subjected to an intravenous injection of Ag₂S superdots and for control mice as obtained 1 and 28 days after injection. (n= 3 for each group). Error bars corresponding to standard error of the mean \pm SEM.

Figures RL9 (a)-(c) show the time evolution in the serum content of ALT, ALP and AST hepatic enzymes for mouse injected with Ag₂S superdot from 24 h to 28 days. The grey region indicates the range of concentrations reported for healthy mice, as obtained from (SAFETEC, GUIA DE ANIMALES DE LABORATORIO I, SEGUNDA EDICION, Rettenmaier iberica JRS). Importantly, none of the hepatic-related enzymes after injection of Ag₂S superdots presented values outside the ranges reported for healthy mouse in the bibliography. Our data suggest that, although our Ag₂S superdots are preferentially accumulated in the liver they do not exert a significant cytotoxic effect during the 28 days after treatment. This data are in accordance to that reported elsewhere for Balb/c injected with 15mg/kg dose of Ag₂S dots. (Y. Zhang, *et al.*, Biomaterials 34.14 (2013): 3639-3646).

Fig RL9 (a)-(c) Time evolution of the ALT, ALP, AST enzyme contents after intravenous injection of Ag₂S superdots, respectively (n= 3 for all groups). Error bars corresponding to standard error of the mean \pm SEM). Gray areas indicate the reference ranges associated to healthy mice.

Figures RL10(a) and (b) show the time evolution of creatinine and bilirubin concentrations after intravenous injection of Ag₂S superdots. Again, the gray area indicates the range of values reported for healthy mice (the levels of creatinine are borderline with the reported range, however this can be partially explained due to the youth of the animals used (1-2 months-old) as well as some differences in weight in animals randomly divided into the same groups. As occurred with hepatic enzymes, the concentrations of both creatinine and bilirubin after intravenous injection of Ag₂S superdots remain within the healthy referenced ranges. These last results are in accordance with the results included in our original version on **Figure 5**.

Fig. RL10. (a)-(b) Time-course of creatinine and total bilirubin concentration after intravenous injection of Ag₂S superdots, respectively (n= 3 for all groups). Error bars corresponding to standard error of the mean \pm SEM). Gray areas indicate the reference ranges associated to healthy mice.

Histological studies have been also conducted. These are described in Supplementary Section S21. Histological images included in **Supplementary Figure 22** reveal the administration of Ag₂S superdots does not cause any lesion to the spleen, liver or kidneys.

Reviewer #3

In this manuscript, the authors have developed ultra-sensitive Ag₂S, which has a strong emission in the NIR-II window, by introducing an additional protective shell layer on the hydrophobic Ag core. The shelling process with femtosecond laser irradiation is simple and can efficiently reduce structural defects resulting in enhanced quantum yield of Ag₂S. Additionally, the physical and optical characterizations of Ag₂S dots before and after ultrafast laser irradiation have done satisfactorily. However, its application toward *in vivo* imaging seems to be very limited, which is not fully supported experimentally.

We appreciate the reviewer's willingness to evaluate and comment on our work. We have addressed all their queries, as shown in the points below.

R3.1. In addition, the benefit of low dose/ low power in the NIR-II imaging is not fully demonstrated.

This is a relevant point, as the advantage of using low irradiation densities and low concentrations of Ag₂S superdots should be clearly stated in the manuscript. Please note that in our original version we included some *in vivo* results showing that the laser-induced heating of live mice during the acquisition of NIR-II fluorescence images strongly depends on the laser power density used (Supplementary Fig. 16 in the revised version). That figure shows how the laser power density used for image acquisition is critical when trying to keep the animal temperature at its basal level. Increasing the temperature above its basal value could cause stress in the animal and, hence, can subtract validity from the obtained results, besides causing unnecessary damage to the animal. The laser power densities used for *in vivo* imaging when Ag₂S superdots are used as contrast agents are so low that the laser-induced temperature increment is lower than 1 degree, so that it can be considered negligible. This demonstrates why the use of low laser power densities is essential for *in vivo* imaging.

We also claim in our work that the use of low administration doses is beneficial. There are several arguments to support this claim. The first one is that in order to develop a cost-effective probe for NIR-II *in vivo* imaging, we have to reduce the amount of material required high contrast. Secondly, the use of low administration doses guarantees the minimization of possible adverse (toxicity) effects. Note that in this work we demonstrate a negligible *in vivo* toxicity of our Ag₂S superdots for the low administration doses required for high contrast *in vivo* imaging.

Finally, it should be noted that Ag₂S superdots show a non-negligible light-to-heat conversion efficiency that results from their relatively low QY. Simple calculations revealed that for the brightest Ag₂S superdots with QY close to 10% the light-to-heat conversion efficiency is above 90%. This means that Ag₂S superdots can contribute to the laser-induced heating during *in vivo* imaging experiments. Thus, the lower is the concentration of Ag₂S superdots the smaller is the laser-induced thermal loading so minimum administration doses are also a requirement to minimize the risk of excessive heating. To illustrate this last issue, we designed a simple experiment where the light-induced heating of Ag₂S superdot dispersions at different concentrations (0.35, 0.25, 0.15 and 0.05 mg/mL) was experimentally determined. **Figure RL11(a)** shows the thermal images of these solutions under simultaneous excitation with a 808 nm laser beam. **Figure RL11(b)** shows the temperature of each dispersion for different laser power densities. The heating effects can be significantly reduced when operating at low concentrations and low power densities. These data and the accompanying discussion have been included in the Supplementary Information of our revised manuscript (Supplementary Fig. 18).

Figure RL11. (a) Thermographic image of Eppendorf tubes containing aqueous dispersions of Ag₂S superdots at different concentrations. (b) Heating efficiency curve obtained for the different dispersions of Ag₂S superdots.

R3.2. Optophysical analyses of Ag₂S dots are well presented in this manuscript. However, critical experimental details are not given. What is the temperature for all assays? What is the exact aqueous buffer or biological media used? How long was incubation? Was serum used? If so, what type and concentration? In the absence of serum, buffered salt solutions alone (i.e., cell culture media or PBS), immediately initiate signal transduction cascades turning on quiescence and/or apoptosis genetic programs.

Following the reviewer's suggestion further experimental details have been included in the revised version of our manuscript. In particular, we have clearly stated that all the assays have been performed at room temperature. Regarding the buffer, all the measurements were performed in PBS. In this revised version, we have also characterized the luminescence properties of Ag₂S superdots in different media including PBS, FBS and DMEM. Results obtained and the corresponding discussion are given in our answer to point 5 of this reviewer.

We also understand from this point that the reviewer has some concerns about the toxicity of our Ag₂S superdots at the cellular level. In addition and with the aim of providing a complete and convincing answer to reviewer as well as to complete the toxicity evaluation of our Ag₂S superdots, we have performed some *in vitro* toxicity assays. In particular the toxicity of Ag₂S superdots has been evaluated in HeLa (human cervical adenocarcinoma), NIH-3T3 (swiss mouse embryo fibroblasts) and IMR-90 (human lung fibroblasts) using the thiazolyl blue tetrazolium bromide (3-(4,5-dimethyl-thiazol-2-yl)-2,5-diphenyltetrazolium bromide, MTT) assay. Briefly, cells in complete medium (DMEM with 10 % FCS, pyruvate, glutamine, penicillin-streptomycin and gentamicin) were seeded in 96-well culture plates at a density of 10⁴ cells/well. After 24 h of incubation (37 °C, 5 % CO₂) nanoparticles were added to cells at different concentrations up to 200 mg/mL and further incubated for 48 h. Then, MTT solution was added for 4 h to the plate and afterwards, MTT reaction was stopped by adding a solution of dimethylformamide-SDS. Finally, the plate was gently shaken for 2 h to dissolve formazan crystals prior to measure 570/590 nm absorbance in an Appliskan (Thermo Scientific) plate reader. Corrected absorbance was transformed to percentage of cell viability using the following formula: cell viability % = (Abs sample/ Abs control)x100, where Abs is the corrected absorbance at 570 nm after subtracting the absorbance at 590 nm. The viability results are included in **Figure RL12** of this response letter. The cytotoxicity of Ag₂S superdots has been found to be relatively low as detailed below.

In the case of the experiments with HeLa cells, the cell viability was as high as 95% even at nanoparticle concentration as large as 200 micrograms/mL. For 3T3 and IMR90 cells, the viability was around 75% for the highest NP concentration studied (200 $\mu\text{g/mL}$). We would like to highlight the harsh experimental conditions used in these experiments in which the concentration of nanoparticles (up to 200 $\mu\text{g/mL}$) and incubation times (48 h) are significantly higher than the ones used in other studies (G. Chen, *et al.* Advanced Functional Materials 24.17 (2014): 2481-2488; C. Wang, *et al.*, Small 8.20 (2012): 3137-3142).

The experimental data included in **Figure RL12** of this response letter together with this discussion has been included in the Supplementary Information of our revised version as Supplementary Fig. 21.

Figure RL12. Cell viability of HeLa, 3T3 and IMR90 cell after 48 h of incubation with different concentration of Ag₂S superdots in the cell medium as determined by MTT assay.

R3.3. Another caveat of this study is poor in vivo assay. Additional studies including PK/PD, such as single dose/repeated dose acute/chronic toxicity, are required to understand the possibility of using Ag2S dots in the clinic. This assay may be beyond the scope of this manuscript, but I recommend performing at least biochemical analysis, biodistribution, and clearance study to prove the potential use of Ag2S in animal models.

The point raised by the review here is the same as one of the issues raised by reviewer 2. Our detailed answer is provided in point 4 of the response to reviewer 2. As described there, we have performed detailed toxicity analyses and histological studies that are included in the revised version of the manuscript.

R3.4. What is the maximum penetration depth of Ag2S super dots in the NIR-II imaging setup? This can be simply performed using a tissue phantom.

Following the reviewer's suggestion, we have designed and performed a simple experiment to determine the penetration depth of Ag₂S superdots in our NIR-II imaging setup. We have filled a quartz cuvette with a solution of Ag₂S superdots and placed a piece of biological tissue (chicken breast) cut to resemble a right

triangle (**Figure RL13(a)**) on top of the cuvette. When illuminating the whole system, a single fluorescence image (such as the one included in **Figure RL13(b)**) allows obtaining the values of luminescence intensity at different depths (such as the one represented in **Figure RL13(c)**). With this set of data, it is possible to estimate the traditional penetration depth, δ_{trad} , and the extrapolated penetration depth, δ_{max} , of the luminescence. To do this, the experimental data are fitted to $I_o + \Delta I e^{-L/\delta_{trad}}$, where I_o is the background intensity detected by the InGaAs camera when the sample is not being excited and ΔI is the signal increment caused by the excited cuvette with no tissue on top. The traditional penetration depth δ_{trad} is then directly obtained with the fitting and corresponds to the tissue thickness that decreases the luminescence intensity down to $I_o + \Delta I/e$. The extrapolated penetration depth, however, is found by the expression $\delta_{max} = \delta_{trad} \ln(\Delta I/\sigma_{noise})$, where σ_{noise} is the intrinsic noise of the measurement. δ_{max} , therefore, corresponds to the highest depth at which a signal greater than the intrinsic noise of the measurement can be detected.

Figure RL13. (a) Optical image of the cuvette with a tissue of variable thickness on top. (b) Fluorescence image obtained when illuminating the system with an 808 nm laser. (c) Fluorescence intensity at different tissue depths. (d) Penetration depth and maximum penetration depths as defined in the text obtained for different laser power densities.

We have also investigated the dependence of the traditional and the extrapolated penetration depths with the excitation power density. The experimentally obtained data are included in **Figure RL13(d)**. It can be observed that δ_{trad} presents a constant value of 3.2 mm while δ_{max} follows a logarithmic trend, increasing from 1 to 2 cm. This set of results demonstrate that, under our experimental conditions, Ag₂S superdots allow for fluorescence imaging into tissues at depths larger than 1 cm. This information is now clearly stated in the revised version and data included in **Figure RL13** has been included in the revised version of Supplementary Information (Supplementary Fig. 17).

R3.5. Two longpass filters (Thorlabs FEL850) were used in this study. For better resolution, NIR-II imaging should be carried out with a longer wavelength filter such as 1100 nm LP.

The reviewer is absolutely correct in pointing out that the spatial resolution of NIR-II fluorescence images could be obtained by adding longpass filters. Previous works have demonstrated that longpass filters with cut-on wavelengths above 1000 nm enable improving the spatial resolution of NIR-II images. This is due to a

better removal of tissue autofluorescence, together with a reduction in the distortions caused by optical scattering (known to be larger as the photon wavelength decreases), as the contribution of short wavelength photons is removed from the images. It is worth saying, however, that in our experimental conditions we expected that the use of 1100 nm long pass filters would have little impact on the final resolution of fluorescence images. Please note that the emission band of Ag₂S superdots is centered at 1200 nm and that the net contribution of photons with wavelength shorter than 1100 nm to the overall intensity has been calculated to be below 5 % (see **Figure RL14**). Thus, the contribution of these highly scattered photons to our fluorescence images can be considered as negligible.

Figure RL14. Schematic diagram of the photon contribution under 1100 nm (red region) to the whole intensity using the Ag₂S superdots.

To test this hypothesis, we have designed a simple experiment where we could infer the improvement in the spatial resolution of fluorescence images by using an additional 1100 nm longpass filter (FEL1100). The idea was to compare the fluorescence images of a thin cuvette filled with a solution of Ag₂S superdots solution in the presence of a tissue of variable thickness. Typical fluorescence images of the thin cuvette are included in **Figure RL15(a)** of this response letter. The spatial resolution of the fluorescence images obtained for different tissue thickness and with and without a longpass 1100 nm filter was estimated by analysing the intensity profiles perpendicularly to the cuvette channel. **Figure RL15(b)** includes some characteristic intensity profiles obtained in absence and presence of the longpass 1100 nm filter and for different tissue thicknesses. The full width at half maximum (FWHM) values of the intensity profiles included in **Figure RL15(b)** have been included in **Figure RL15(c)**. The greater the difference of FWHM values to the original FWHM (0.4 mm, measured with no tissue on top), the less spatial resolution that image would have. According to **Figure RL15(c)**, for injection depths of 3 mm or lower, the use of a 1100 nm longpass filter does not lead to any significant improvement in spatial resolution. For deeper injection sites, however, one might have to be more aware of the differences and prioritize the use of FEL1100. For this reason, we have used FEL1100 in the new *in vivo* experiments included in the updated version of the manuscript.

Figure RL15. (a) Fluorescence images of a cuvette filled with a dispersion of Ag_2S superdots illuminated by an 808 nm laser. The dashed rectangle on the right corresponds to the region where a piece of tissue was placed in front of the cuvette. **(b)** X-Profiles of fluorescence intensity after passing through different depths of tissue. **(c)** Comparison of FWHM values as obtained for different combinations of longpass filters.

R3.6. This study is missing an important control. ICG will be a good model, although it uses a tail of emission for NIR-II imaging. Comparison should be done in serum containing media.

We agree with the reviewer that when introducing a new NIR-II luminescent nanoprobe, the comparison of their performance with that of ICG is mandatory. We would like to note that such comparison was already in our original version. Supplementary Fig. 10 in the former Supplementary material (Supplementary Fig. 14 in the revised version) compares the NIR-II fluorescence images obtained by our system of aqueous dispersions of ICG and Ag_2S superdots. The main conclusion extracted from these images is that, in our experimental conditions and with our detecting camera, the emission of ICG was hardly detectable, while the Ag_2S superdots provided a clear signal. This is why we decided not to include *in vivo* images of mice injected with ICG. Note that this decision was adopted trying to minimize the number of animals used in our study taking into account the minimal chances of registering any signal generated from intravenously injected ICG. At this point we should note that there are different works reporting on nice NIR-II fluorescence *in vivo* images by using ICG as contrast agent. We think that in those works the detection system used was of larger sensitivity than ours. For instance in the work published by R. Bhavane et al (Scientific Reports 8, 14455 (2018)) the *in vivo* images of a mouse after intravenous injection of ICG were obtained with an InGaAs camera cooled down to $-80\text{ }^\circ\text{C}$. The images included in our work were obtained with an InGaAs camera cooled down only to $-40\text{ }^\circ\text{C}$. The more intense background noise level of our camera might be responsible for the poor detection of the signal generated by ICG. Note that, the improvement demonstrated in our paper could be even more pronounced when using a better (lower noise) infrared camera such as those used by other groups working in NIR-II *in vivo* imaging. This fact is clearly stated in our revised version.

In any case, and following the suggestion raised by the reviewer, we have performed additional experiments comparing the fluorescence images and spectra of Ag_2S dots, Ag_2S superdots and ICG in different solvents (PBS, FBS and DMEM). Results are included in **Figure RL16** of this response letter. As can be observed in all the cases the NIR-II brightness provided by Ag_2S superdots is superior. Furthermore, the different solvents do not affect their spectral properties as the same NIR-II emission spectra has been obtained in all the solvents. This information is included in the revised version (Supplementary Fig. 15).

Figure RL16. Optical and fluorescence images of Eppendorf tubes filled with either a control solution or dispersions of ICG, Ag₂S dots or Ag₂S superdots in (a) PBS, (b) FBS and (c) DMEM. Fluorescence intensity as measured by hyperspectral imaging of a control solution or solutions of ICG, Ag₂S dots or Ag₂S superdots in (d) PBS, (e) FBS and (f) DMEM.

R3.7. Page 3, “The combination of already available Ag₂S dots and current image acquisition technologies allows for imaging depths of few centimetres whereas clinical applications require penetration depths one order of magnitude larger.” This sentence needs to be supported by reference(s). As far as I know, the penetration depth of NIR-II imaging is limited to a few millimeters, not centimeters in the body. Although this is argued in Introduction, no supporting data is presented to support this statement as well.

We agree with the reviewer. In order to avoid any misunderstanding these sentences have been removed from the introduction. In the revised version we have just indicated that the translation into the clinics of NIR-II fluorescence imaging requires the improvement in the fluorescent properties of currently available NIR-II probes in order to increase the maximum penetration depth of the obtained images.

R3.8. Page 3, “we have developed a novel methodology that permits a 80-fold increment in the quantum yield of Ag₂S dots.” Although the chemical reaction for NPs is the same, the methodology of laser irradiation is interesting. I was wondering if this photo radiation induced shelling method could be applied to the other nanocrystals to enhance their QY.

This is a very interesting question also raised by reviewer 1 (see point R1.1). We are firmly convinced that this method can also be applied to other systems. Note that the method is based on the laser-induced growth of a protective layer of AgCl, being the Ag atoms coming from Ag NPs present in the solution and Cl atoms coming from the solvent (chloroform). So, in principle, the method could also work for other NPs dispersed in chloroform in presence of Ag NPs. Of course, the reactivity of the surface of different nanoparticles is a factor that could also play an important role in the process. In the particular case of Ag₂S dots the high reactivity of their surface promote the formation of the AgCl protective layer. It is not clear whether or not the method will also work for other NPs with different surface chemistry, but it is definitely worth investigating, as we are currently doing in our group.

Reviewers' comments:

Reviewer #1 (Remarks to the Author):

The revised version has answered all the concerns raised by the reviewers, so that it is ready for publication now.

Reviewer #2 (Remarks to the Author):

Revision is satisfactory and can be accepted now.

Reviewer #3 (Remarks to the Author):

The authors did not address the reviewers' comments well, and there are still serious problems with some biological data, as outlined below. The presented results are lacking confidence and relevance; therefore, I do not believe this manuscript is suitable for publication in Nature Communications.

1. The authors newly added several *in vivo* imaging results. But overall the image quality is very poor compared to the previously reported results from other groups (Nat Commun 2018, 9, 1171, ACS Nano 2019, 13, 248, Nat Commun 2019, 10, 1058, Adv Mater 2018, e1802546), which are not publishable at all. If it's really "superdot," why the *in vivo* images are so poor? Especially, it's very hard to distinguish the signal from the background even in superdots images shown in Figure 4f. The author should describe the analytical details in the pixel intensity profiling. It does not seem to match with NIR images. Comparison between Ag₂S dots and Ag₂S superdots in Fig 4f,g seems to be unfair due to the difference in anatomy of individual mice.
2. All references related on NIR-II imaging are outdated and missing a lot of important papers including Nature Biomedical Engineering 2017, 1, 0010. The groups at Stanford University and NIBIB are leading this field, and many important papers have recently been published in Nature sister journals.
3. The author should assess a maximum tolerated dose for Ag₂S superdots if the minimum dose requirement of Ag₂S superdot is a real benefit. If it shows any toxicity at such a low dose it is not useful. The authors stated that 0.5mg/kg dose was used for most *in vivo* imaging, but indeed 5mg/kg was used for biodistribution study, which is over 10-fold higher than regular ICG dose in the clinic. Additionally authors should explain why infusion via right jugular vein was used, which is not common.
4. H&E results seems to be quite different in Ag₂S superdots images compared with the control at 1-day post-injection.
5. The authors newly conduct to confirm the spatial resolution with different long-pass filters (850 and 1100 nm) and stated "In vivo images were also acquired using an additional 1100 nm long-pass filter, following previous works revealing that the use of such filter could increase the resolution of the NIR-II fluorescence images. In our case, as it is described in Supplementary Section S23, the use of such filter did not improve the image resolution." But I don't agree with the conclusion on the spatial resolution. More systematic experiments are necessary to make a concrete conclusion.
6. There are a lot of typos, grammatical and formatting errors in the whole manuscript and supporting info.

LIST OF CHANGES

1.- Quality of the NIR-II fluorescence images reported respective to those previous published.

We have considered the point raised by the reviewer regarding the quality of our *in vivo* NIR-II fluorescence images compared to those previously reported by other groups. We have included a new figure in the Supplementary Information (**Supplementary Figure 18**) summarizing the most representative literature on NIR-II *in vivo* image of vessels, including those reported in the references suggested by the reviewer in their last report (see **Figure RL1** of this response letter). When comparing the fluorescence images included in **Figure RL1** to those in our manuscript (such as the ones of **Figure 4f-h**) we do not believe that the quality of the latter can be described as “poor”. While some of the *in vivo* NIR-II fluorescence images reported by other research groups seem to have better resolution and/or contrast than ours, this can be explained considering the differences in experimental conditions, particularly in camera specifications. These are summarized in **Table RL1** of this response letter and also in the Supplementary Information of the revised version of our manuscript (**Supplementary Table 3**). Most of the fluorescence images in **Figure RL1** were obtained with higher-end NIR-II cameras that have better technical performance than ours. The majority of published works in this field report using deep-cooled (< -80 °C) InGaAs cameras that present a substantially lower dark noise (see **Table RL1**) than ours, operating at -40 °C. In addition, the pixel size of our camera is 30x30 μm^2 , which is also larger than that of the cameras used by other groups (typically 20x20 μm^2). A larger pixel size translates to a lower spatial resolution. All this considered, the differences in quality between our NIR-II images and those obtained by other groups can be explained by the worse performance of our camera. This is clearly stated in the revised Supplementary Information.

As stated earlier, the novelty of our manuscript lies in the approach to drastically improve the fluorescence properties of NIR-II probes. The *in vivo* images are included in our manuscript to prove that this improvement directly impacts the contrast and quality of the images. That is the reason why we compare *in vivo* images obtained using Ag₂S dots and superdots under identical same experimental conditions, as comparing our images with those obtained by other groups is unfeasible due to the use of different detection systems.

Figure RL1: Representative NIR-II *in vivo* fluorescence images of mouse vasculature as obtained by different research groups using different NIR-II probes and detection systems that are summarized in Table RL1 of this response letter. (a), (b), (c), (d), (e) and (f) were obtained from Y. Li et al, *ACS nano* **13**, 248-259 (2019), S. Wang et al, *Nat Commun* **10**, 1-11 (2019), H. Wan et al, *Nat Commun* **9**, 1171 (2018), S. Zhu et al. *Adv Mater* **30**, 1802546 (2018), C. Li et al, *Biomaterials* **35**, 393-400 (2014), G. Hong et al, *Nat Med* **18**, 1841-1846 (2012).

Probe	Injected dose	P_{laser} (mW cm ⁻²)	Camera specifications				Ref.
			Model	Noise	Pixel size (μm ²)	Operating T (°C)	
p-FE	200 μL of an OD 6.5 (at 808 nm) solution	70	Princeton Instruments 2D OMA-V, USA	5000 e/p/sec	30x30	-100	(1)
NaLnF ₄ : Gd,Yb,Er,Ce	200 μL, 3 mg mL ⁻¹	100	NIRvana, Princeton Instruments	10 e/p/sec	20x20	-80	(2)
Cyanine dye	200 μL, 1.25 nmol	150	NIRvana, Princeton Instruments	150 e/p/sec	20x20	-85	(3)
Cyanine dye	30-150 μg	150	Princeton Instruments	Un-specified	Un-specified	Un-specified	(4)
Ag ₂ S dots	50 μL, 1 mg mL ⁻¹	45	Photonic Science, UK	40 e/sec	15x15	-25	(5)
SWNTs	200 μL, 0.1 mg mL ⁻¹	140	LN cooled NIRvana, Princeton Instruments	10 e/p/sec	20x20	-190	(6)
Ag ₂ S dots	100 μL, 0.15 mg mL ⁻¹	40	Xeva 320 Xenics	10000 e/sec	30x30	-40	This work

Table RL1: Experimental conditions used for the acquisition of NIR-II *in vivo* fluorescence images of vasculature included in Figure RL1. (1) H. Wan et al, *Nat Commun* **9**, 1171 (2018), (2) Y. Li et al, *ACS nano* **13**, 248-259 (2019), (3) S. Wang et al, *Nat Commun* **10**, 1-11 (2019), (4) S. Zhu et al. *Adv Mater* **30**, 1802546 (2018), (5) C. Li et al, *Biomaterials* **35**, 393-400 (2014), (6) G. Hong et al, *Nat Med* **18**, 1841-1846 (2012).

2.- Quantitative analysis and fair comparison between the NIR-II *in vivo* fluorescence images obtained with Ag₂S dots and superdots.

In their last report, reviewer #3 mentioned the need for a fair comparison between the NIR-II *in vivo* images obtained by Ag₂S dots and superdots. To provide readers with a fairer comparison, we have re-analysed the NIR-II fluorescence images with a more systematic approach. In the revised version of our manuscript, we now include the NIR-II fluorescence images obtained before and after Ag₂S injection (see **Figure RL2a**). As seen in this image, a signal originating from NIR-II tissue autofluorescence is present in the NIR-II image of the limb before injection. Hereafter, these pre-injection images will be considered as the background images. **Figure RL2a** also shows the NIR-II images of mice limbs 15 seconds after intravenous administration of either dots or superdots. The presence of either contrast agent in the bloodstream leads to an increase in the overall fluorescence intensity, which is highest at the blood vessels. The fluorescence images obtained after dot or superdot administration contain the emission generated by the contrast agents and the background signal. Therefore, post-injection images are considered as the “signal+background” images. The “signal” images (i.e. NIR-II images where only the contribution of the dots or superdots is present, as shown in **Figure RL2b**) are obtained by subtracting the pre-injection images to the post-injection ones to remove the autofluorescence background. **Figure RL2b** evidences that using Ag₂S superdots instead of dots increases image contrast and allows visualization of secondary blood vessels. The signal-to-background images, shown in **Figure RL2c** can then be constructed dividing the signal and background images. The signal-to-background ratio is clearly superior in the case of superdots. To quantify the improvement, we have analysed the pixel profiles across the saphenous artery, as indicated by a dashed line in **Figures RL2b and c**. **Figure RL2d** shows the pixel profiles obtained from the signal images in **Figure RL2b**. The higher brightness of superdots leads to an enhancement in the NIR-II signal at the femoral artery that is close to 60 %. **Figure RLc** shows the same pixel profiles obtained from the signal-to-background images in **Figure RL2c**. Again, the superdots show a superior performance, leading to an improvement in the signal-to-background ratio close to 90 %.

Figure RL2: NIR-II *in vivo* imaging performance of Ag₂S dots and superdots (a) NIR-II fluorescence images of the right hind limbs of two mice immediately before (top) and 15 seconds after (bottom) of an intravenous injection of Ag₂S dots (left) or superdots (right). (b) Net intensity images obtained from subtracting the background images (top row in a) from the signal images (bottom row in a). (c) Signal-to-background images obtained by dividing the net intensity images in b by the background images (top row in a). (d) Net intensity and (e) signal-to-background ratio obtained along a line profile crossing the saphenous artery (indicated as dashed white lines in b and c).

To clarify this point, we have added the data in Figure RL2 to Figure 4 in the manuscript (Figure 4f-j) and the following paragraph to our revised text:

“Ag₂S superdots enable improved *in vivo* visualization of blood vessels, as shown in Fig. 4f-j, where we compare the images of the right hind limbs of two mice after intravenous injection of either dots or superdots (see Methods). Fig.4f shows the magnified luminescence images of the limb immediately before (top) and 15 s after injection (bottom). Although the pre-injection background (caused by tissue autofluorescence) was similar in both cases, the image contrast after injection is substantially higher after injection of superdots (left) than in the case of dots (right). We quantified the improvement in image contrast by calculating the signal-to-background ratio for both sets of images. This was done by subtracting the background from the overall emission to obtain net signal images (Fig. 4g), which were then divided by the background to calculate the signal-to-background ratios (Fig. 4h). Both the net signal (Fig. 4i) and the signal-to-background (Fig. 4j) line profiles along the femoral artery (dashed lines in Fig. 4g-h), shown indicated a superior performance of the Ag₂S superdots as compared to dots.

From the pixel profiles in Fig. 4i-j indicate that Ag₂S superdots improve the net signal by 60% and the signal-to-background ratio by 90% when compared to conventional Ag₂S dots. Further discussion on the NIR-II images, focusing on the relative quality of the ones presented here to those shown in previous work by other groups, is included in **Supplementary Section 18.**"

3.- Further discussion on enzyme level results and toning down the claims regarding the potential clinical application of the Ag₂S superdots.

The data on the effect Ag₂S superdots on hepatic enzyme levels have been reanalysed and discussed in greater depth. As seen in **Supplementary Figure 19**, all hepatic enzymes levels fall within the normal ranges for mice as provided by the reference laboratory. These ranges are relative broad as they are obtained from mice of different strains, ages and genders, all of which affect the expression levels of these enzymes to some extent. Then, the fact that the enzyme levels obtained were within these normal ranges does not guarantee the absence of liver damage. Nevertheless, we would like to highlight that, as shown in **Supplementary Figure 19a**, the ALT levels remained stable for the duration of the experiment, which is particularly relevant. Although ALT, ALP and AST are biomarkers of liver damage, only ALT is found primarily in the liver. ALP is found in the liver and bones whereas AST is found in the liver, kidneys, brain, heart and skeletal muscle. Therefore, it is accepted that preliminary detection of hepatocellular injuries can be elucidated by just monitoring alterations in the ALT enzyme (see "Towards better diagnostic tools for liver injury in low-income and middle income countries" by Saundria Moed, and Muhammad H Zaman in *BMJ Global Health* (2019), 001704, doi:10.1136/bmjgh-2019-001704). Therefore, the stability of ALT levels can be considered an indicator of absence of relevant hepatic damage. While **Supplementary Fig. 19 b, c** reveal changes in the ALP and AST levels, the measured values still fall within normal ranges. The observed alterations in these enzymes can be tentatively explained considering that the stress induced to the animals due to their manipulation during the treatment affects the hepatic metabolism and thus its enzymatic levels. In addition, **Supplementary Fig. 20a** shows that no elevations occur in creatinine that would have suggested an impaired function of the kidneys. The stability in bilirubin values shown in **Supplementary Fig. 20b** confirms the absence of liver damage and of any significant destruction of erythrocytes, as bilirubin is a hemoglobin metabolite excreted via the bile. All these results are consistent with the histopathological studies, in which we did not observe any sign of inflammation or any structural or cytological alterations of the liver, kidney or spleen.

Both the *ex vivo* toxicity assays and the analysis of enzyme levels seems to indicate a minimum toxicity for the Ag₂S superdots. Nevertheless, further toxicity assays must be performed before considering the clinical application of Ag₂S superdots. These should include the study of the maximum tolerated dose, clearance pathways, and a full assessment the effect of these nanomaterials on the physiology, metabolism, behavior and cognition. Such evaluation of the *in vivo* biocompatibility of Ag₂S superdots implies a large volume of experiments that is out of the scope of this work.

In the absence of such complete characterization, we cannot claim that Ag₂S superdots are candidates for immediate clinical application. In the revised version, we clearly state that the translation of Ag₂S superdots as NIR-II optical probes from the preclinical to the clinical level is not immediate and will require further, in-depth work.

4.- Literature updating.

The following references have been included in this revised version:

- Hong G, Antaris AL, Dai H. Near-infrared fluorophores for biomedical imaging. *Nat Biomed Eng* **1**, 0010 (2017).
- Cao J, *et al.* Recent Progress in NIR-II Contrast Agent for Biological Imaging. *Frontiers in Bioengineering and Biotechnology* **7**, (2019).
- Tian R, *et al.* Multiplexed NIR-II Probes for Lymph Node-Invaded Cancer Detection and Imaging-Guided Surgery. *Adv Mater*, 1907365 (2020).
- Li Y, Zeng S, Hao J. Non-invasive optical guided tumor metastasis/vessel imaging by using lanthanide nanoprobe with enhanced down-shifting emission beyond 1500 nm. *ACS nano* **13**, 248-259 (2019).
- Wang S, *et al.* Anti-quenching NIR-II molecular fluorophores for in vivo high-contrast imaging and pH sensing. *Nat Commun* **10**, 1-11 (2019).
- Wan H, *et al.* A bright organic NIR-II nanofluorophore for three-dimensional imaging into biological tissues. *Nat Commun* **9**, 1171 (2018).
- Li L, *et al.* A Rationally Designed Semiconducting Polymer Brush for NIR-II Imaging-Guided Light-Triggered Remote Control of CRISPR/Cas9 Genome Editing. *Adv Mater* **31**, 1901187 (2019).
- Tian R, *et al.* Rational design of a super-contrast NIR-II fluorophore affords high-performance NIR-II molecular imaging guided microsurgery. *Chemical science* **10**, 326-332 (2019).
- Das P, Santos S, Park GK, Hoseok I, Choi HS. Real-Time Fluorescence Imaging in Thoracic Surgery. *The Korean journal of thoracic and cardiovascular surgery* **52**, 205 (2019).
- Li C, *et al.* In vivo real-time visualization of tissue blood flow and angiogenesis using Ag2S quantum dots in the NIR-II window. *Biomaterials* **35**, 393-400 (2014).
- Hong G, *et al.* Multifunctional in vivo vascular imaging using near-infrared II fluorescence. *Nat Med* **18**, 1841-1846 (2012).
- Zhu S, *et al.* Repurposing Cyanine NIR-I Dyes Accelerates Clinical Translation of Near-Infrared-II (NIR-II) Bioimaging. *Adv Mater* **30**, 1802546 (2018).

REVIEWERS' COMMENTS:

Reviewer #3 (Remarks to the Author):

The manuscript represents an important step forward to the design of nanoparticle-based superbright NIR-II imaging probes for noninvasive intraoperative imaging. This idea is innovative, and the experimental data are sufficient to support the conclusions. The authors also addressed the reviewer's concerns fairly adequately, thus this manuscript should be acceptable for publication in Nat Commun.